

# Towards variational retrieval of warm rain from passive microwave observations

David Ian Duncan[1], Christian D. Kummerow[2], Brenda Dolan[2], and Veljko Petković[2]

[1]Department of Earth, Space, and Environment, Chalmers University of Technology, SE-41296, Gothenburg, Sweden
[2]Department of Atmospheric Science, Colorado State University, Fort Collins, CO

**Correspondence:** David Duncan (david.duncan@chalmers.se)

**Abstract.** An experimental retrieval of oceanic warm rain is presented, extending a previous variational algorithm to provide a suite of retrieved variables spanning non-raining through predominantly warm raining conditions. The warm rain retrieval is underpinned by hydrometeor covariances and drizzle onset data derived from CloudSat. Radiative transfer modelling and analysis of drop size variability from disdrometer observations permit state-dependent observation error covariances that scale
with columnar rainwater during iteration. The state-dependent errors and nuanced treatment of drop distributions in precipitating regions are novel and may be applicable for future retrievals and all-sky data assimilation methods. This retrieval method can effectively increase passive microwave sensors' sensitivity to light rainfall that might otherwise be missed.

Comparisons with space-borne and ground radar estimates are provided as a proof of concept, demonstrating that a passive-only variational retrieval can be sufficiently constrained from non-raining through warm rain conditions. Significant deviations
from forward model assumptions cause non-convergence, usually a result of scattering hydrometeors above the freezing level. However, for cases with liquid-only precipitation, this retrieval displays greater sensitivity than a benchmark operational retrieval. Analysis against passive and active products from the Global Precipitation Measurement (GPM) satellite shows substantial discrepancies in precipitation frequency, with the experimental retrieval observing more frequent light rain. This approach may be complementary to other precipitation retrievals, and its potential synergy with the operational passive GPM retrieval is
briefly explored. There are also implications for data assimilation, as all 13 channels on the GPM Microwave Imager (GMI) are simulated over ocean with fidelity in warm raining conditions.

## 1  Introduction

Global observation of precipitation depends heavily on passive measurements of hydrometeors at microwave wavelengths. Active sensors possess certain advantages relative to passive sensors, but a full global picture of precipitation is currently im-
possible from active sensors alone as they yield limited spatial coverage and may miss near-surface precipitation due to ground clutter effects. While ground radar networks cover some landmasses, a satellite platform is necessary for global observation of rainfall. Accurate observation of the hydrologic cycle at a high spatiotemporal resolution is a worthy goal (Hou et al., 2014), and a task that realistically requires passive microwave rainfall retrievals.



Retrieval of precipitation from passive microwave observations is an under-constrained problem (Stephens and Kummerow, 2007). This is due to many factors, including unknown distributions of ice, mixed phase, and liquid hydrometeors, as well as their horizontal distribution within the sensor field of view (FOV), coupled with limited channels which possess non-independent information content. In effect, there are more unknowns than pieces of independent information, and thus many

assumptions are necessary to make the problem tractable. This has historically been done via algorithms built on empirical relationships (Hilburn and Wentz, 2008; Wilheit and Chang, 1980) or algorithms based on Bayesian principles with Gaussian-distributed parameters (Bauer and Schluessel, 1993; Iturbide-Sanchez et al., 2011; Kummerow et al., 2015), of which variational (VAR) methods form a subset (Rodgers, 2000).

The patterns and magnitude of precipitation over much of the tropical oceans are largely agreed upon, a result of the co-

ordinated study of tropical precipitation from the Tropical Rainfall Measuring Mission (TRMM), which launched in 1997 (Kummerow et al., 2000). In contrast, the stratocumulus regions and high latitude oceans remain areas of disagreement between different observing platforms and among global models (Behrangi et al., 2012; Rapp et al., 2013; Behrangi et al., 2016; Stephens et al., 2010). The launch of the Global Precipitation Measurement (GPM) core observatory in 2014 (Hou et al., 2014) increased the observational capability of sensing global precipitation, but since the Dual-frequency Precipitation Radar (DPR)

has limited sensitivity to the light precipitation so prevalent at high latitudes, uncertainty remains (Skofronick-Jackson et al., 2017). In theory, a variational passive retrieval is sensitive to rainfall below the detectability threshold of DPR, and is also not susceptible to ground clutter that may obscure shallow clouds and precipitation (Liu et al., 2016). Thus a passive-only algorithm may be better suited to retrieval of the light rain rates that are characteristic of high latitude oceans and stratocumulus regions.

GPM's operational passive algorithm, the Goddard Profiling (GPROF) algorithm (Kummerow et al., 2015), leverages the synergy of co-located radar and radiometer observations from GPM to calculate the precipitation rate expectation value for all GPM constellation radiometers. The Bayesian scheme uses the brightness temperature ($T_B$) vector to find an average set of atmospheric profiles that match what the radar would have seen, based on the a priori database. While highly versatile, one weakness of this approach is that it misses hydrometeors below the detectability threshold of the radar, even if the $T_B$s exhibit

signal where the radar does not (GPM Science Team, 2017). Further, while this type of approach gives a satisfactory average answer, it does not explicitly model radiation coming from the surface and atmosphere, blunting the measurements' effective signal to noise ratio by including many surface states and cloud types in the Bayesian average (Duncan et al., 2017).

Warm rain—precipitation driven primarily by collision-coalescence below the freezing level—is particularly challenging to sense from satellite platforms. Passive microwave algorithms are built to exploit the differential signals of emission from liquid

drops and scattering from large drops and mixed phase or frozen hydrometeors, but in the absence of significant emission or scattering, the signal may be from cloud alone or a combination of factors (Stephens and Kummerow, 2007). In spite of these challenges, warm rain is not an insignificant player in the global hydrologic cycle. Warm rain constitutes a majority of precipitating clouds in stratocumulus regions (Lebsock and L'Ecuyer, 2011; Mülmenstädt et al., 2015) and 20% of total rainfall over the Tropical oceans is from warm clouds (Liu and Zipser, 2009). While not missed entirely by current passive retrievals,

some of this emission signal may be missed or misattributed due to its relative subtlety.





The operational data assimilation (DA) community is also invested in passive microwave radiances in precipitating conditions. Successful assimilation of "all-sky" radiances from microwave radiometers can yield a more accurate analysis state from which numerical weather prediction (NWP) models can run (Geer et al., 2017). However, the same factors that cause the retrieval problem to be under-constrained are also relevant for DA schemes (Wang et al., 2012). Thus, microwave radiances

from raining or cloudy pixels are often not included in the data assimilation. If radiances are included, they are accompanied by large observation errors (Lean et al., 2017), diminishing the information content added to the analysis state. Assimilation of satellite radiances is typically done with prescribed and uncorrelated errors—a poor assumption for nearby frequencies especially—although there has been movement towards including correlated observation errors (Bormann et al., 2011, 2016; Weston et al., 2014).

Variational methods for retrievals and DA schemes alike should include realistic estimates of the errors for both the a priori state and observation vector. Whereas prior knowledge from model data or observations can inform a priori error covariances, error covariances applied to the observation vector are more complex, as they should include instrument noise, forward modelling error, and also forward model parameter error as explored by Duncan and Kummerow (2016). For a raining retrieval, the assumption of a drop size distribution (DSD) is a large source of error for the forward model but difficult to quantify

because the true DSD is almost never known. This is effectively a forward model parameter error, assuming that the DSD is not retrieved. As shown by Lebsock and L'Ecuyer (2011), choosing an inappropriate DSD can greatly impact the results of a retrieval, as variations in drizzle rates over ocean are largely explained by variations in drop number concentrations (Comstock et al., 2004). Unfortunately, the distribution of drops in the forward model significantly affects the resultant rain rate and has an effect on the $T_B$ vector, but is not readily retrievable from a single sensor (Mace et al., 2016).

To be clear, variational precipitation retrieval is a very difficult problem to solve for all conditions. This is implicit in the empirical estimate of rain rate in Iturbide-Sanchez et al. (2011) or how CloudSat has no variational retrieval that spans all precipitation types. To make the problem tractable, here we limit the problem to the most straightforward extension to a non-raining retrieval over ocean—that of warm rain. To combat the underconstrained nature of these retrievals, the experimental retrieval described herein is applied to the GMI sensor. GMI possesses lower frequency imager channels and four higher

frequency channels more sensitive to scattering from smaller particles, providing the best information content for sensing precipitation of any extant passive sensor. Additionally, GMI is a good testbed sensor in that it is well calibrated (Draper et al., 2015) and co-locations with DPR are readily available for analysis.

This study builds upon the ocean algorithm developed for the GMI described by Duncan and Kummerow (2016), the Colorado State University 1D variational algorithm (CSU 1DVAR), with several augmentations to extend its applicability into

warm raining conditions. The satellite instruments and datasets used in this study are detailed next. Section 3 addresses three key impediments to a variational precipitation retrieval and offers solutions. Section 4 describes the experimental algorithm's innovations that permit retrieval of warm rain. Section 5 presents a few case studies of GMI overpasses compared against independent rainfall estimates from space-borne and ground radars; statistical analysis comparing 1DVAR rainfall frequency with DPR is also given. Section 6 provides a discussion of limitations, sensitivities, and implications of the retrieval, and the

paper closes with a brief summary and conclusions.



## 2   Data

The GPM core observatory holds two instruments: the GPM Microwave Imager (GMI) and the Dual-frequency Precipitation Radar (DPR). GPM is in a non-Sun synchronous orbit at an inclination of 65° and was launched in February 2014. Compared to its predecessor, TRMM, the higher inclination orbit allows for observation of latitudes well outside the Tropics. GMI is a 13

channel passive microwave radiometer containing channels from 10 to 183 GHz at horizontal (H) and vertical (V) polarisations (Draper et al., 2015). All 13 channels are used in the algorithm described, with $T_B$s coming from the co-registered L1CR product. DPR is a dual-frequency precipitation profiling radar observing at Ku (13.6 GHz) and Ka (35 GHz) bands with a 12 dBZ sensitivity threshold. This study uses GPM V05 brightness temperatures and level 2 products. Both the normal scan (NS) Ku-band only and matched scan (MS) Ku- and Ka-bands combined products are used in this study.

The CloudSat mission's payload is a 94 GHz cloud profiling radar (Stephens et al., 2002). CloudSat was launched in 2007 and flies in the A-Train constellation (L'Ecuyer and Jiang, 2010). At a higher frequency than DPR and with greater radar sensitivity, CloudSat is sensitive to clouds and light rain not seen by DPR, though its signal can attenuate in moderate to heavy precipitation. CloudSat's small footprint permits highly limited spatial sampling. For light precipitation, CloudSat provides the best observational record currently available from satellite, and is thus complementary to GPM observations. CloudSat's

overpasses coincident with GPM were determined using the CloudSat-GPM Coincidence Dataset version 1C (Turk, 2016).

The warm rain retrieval from CloudSat (Lebsock and L'Ecuyer, 2011) is leveraged to construct a priori states usable by a variational retrieval. This algorithm and the associated data product, 2C-Rain-Profile, yields profiles of rain water content, cloud water content, and precipitating ice water content as well as surface rain rate. 2C-Rain-Profile uses a variational approach to match observed radar reflectivities with a two-stream forward model that includes multiple scattering. It employs a

variable DSD chosen specifically for its applicability to warm rain scenes that are dominated by small drops. CloudSat's single frequency radar is supplemented by visible optical depth information from another A-Train sensor to constrain the retrieval of cloud water path.

The GPM Ground Validation team collects data from certain NEXRAD (Next Generation Radar) sites matched with GPM overpasses (GPM Science Team, 2015). NEXRAD operates a dual-pol radar site on the island of Middleton, Alaska at 59°N.

This radar site is ideal for comparisons due to its essentially oceanic location at a latitude frequently sampled by GPM. This ground radar will be referred to as PAIH, its station identifier, hereafter. Ground radar rain rates used in the analysis are from the polarimetric Z-R algorithm (Bringi et al., 2004).

## 3   Impediments

The main impediments to variational retrieval of precipitation over ocean from passive microwave observations can be distilled

down to three factors. In this section, the key impediments to a successful retrieval are enumerated, described, and given solutions. Each is directly tied to an element of the retrieval as described in the following section.

First, it is difficult to differentiate between cloud and drizzle drops from radiances alone, necessitating an assumed partition between cloud water and rainwater in the absence of significant scattering. Second, passive radiances at typical imager frequen-



cies contain almost no information on the vertical structure of hydrometeors. Third, the $T_B$s do not contain enough information to solve for the DSD parameters, but the scattering properties, fall speed, and resultant rain rate of hydrometeors are dependent upon their size distribution. Importantly, the impact on radiances caused by the hydrometeors' distribution depends on the mass of hydrometeors in the atmospheric column.

## 3.1 Partitioning non-scattering liquid

At typical imager wavelengths, cloud droplets lie well within the Rayleigh scattering regime, being instead good emitters of radiation due to their dielectric properties. Mie theory dictates that scattering is proportional to the size parameter ($x = 2\pi r/\lambda$) to the fourth power for a given radius $r$ and wavelength $\lambda$. Even for an effective radius of $100\,\mu\text{m}$, thought to exist in typical drizzle clouds, the size parameter $x$ is equal to 0.19 at $89\,\text{GHz}$, just on the verge between the Rayleigh and Mie scattering regimes. Thus for many drizzle cases, the actual radiometric observations at GMI frequencies will not diverge significantly from simulated observations that neglect scattering.

A simple absorbing/emitting forward model can be run due to the lack of scattering from cloud and drizzle drops. In fact, the predominant lack of scattering from drizzle holds for pristine and polluted regimes, as cloud top effective radii are usually less than $30\,\mu\text{m}$ even for precipitating clouds (Lebsock et al., 2008). However, because non-raining and raining clouds exhibit similar signals, this requires an assumption of partitioning between cloud and rain water emission from passive microwave algorithms. In contrast, a radar algorithm such as that used by CloudSat is more skillful at differentiating between cloud and rain drops because radar backscatter is very sensitive to drop size.

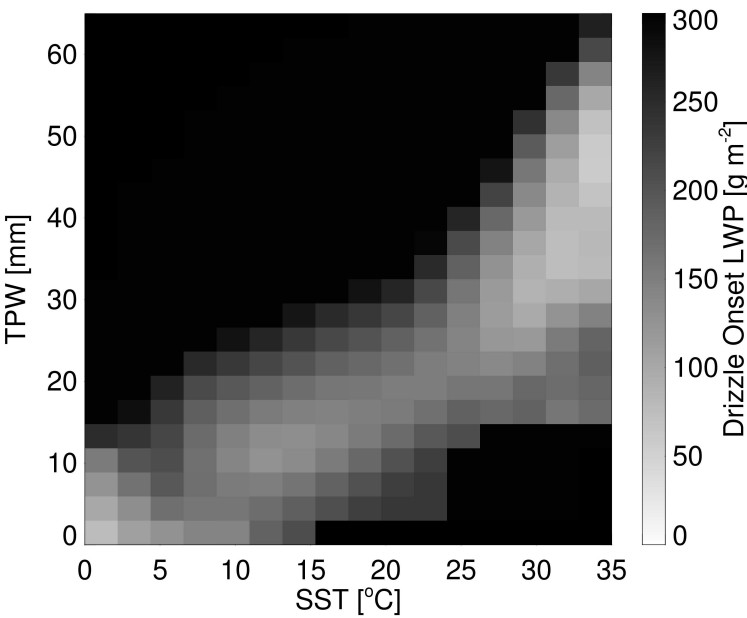

**Figure 1.** Drizzle onset value of LWP, separated by SST and TPW. Regimes with no data are assigned the maximum value, $300\,\text{g\,m}^{-2}$.



To calculate regime-dependent values for the onset of drizzle from liquid clouds, CloudSat data are used. Precipitation frequency observed by CloudSat was analysed and compared to the distribution of LWP as retrieved by the CSU 1DVAR non-raining retrieval for GMI. CloudSat data were averaged to approximate the GMI field of view (FOV). The non-raining 1DVAR retrievals that exhibited very poor fits to observations were assumed precipitating, and the retrievals with high LWP were

5 designated precipitating until the precipitation frequency matched the CloudSat-derived results in each total precipitable water (TPW) and sea surface temperature (SST) regime, effectively ensuring that precipitation frequency mirrors that of CloudSat. This approach implicitly assumes that clouds with higher LWP are more likely to be precipitating, an assumption broadly true in studies of A-Train data (Chen et al., 2011; L'Ecuyer et al., 2009; Stephens and Haynes, 2007). Figure 1 shows the drizzle onset values of liquid water path (LWP) used in this study, subset by TPW and SST. These drizzle onset values are in general

10 agreement with some in the literature (Chen et al., 2011; Lebsock et al., 2008; Mülmenstädt et al., 2015; Wentz and Spencer, 1998) and lower than some others (Iturbide-Sanchez et al., 2011; Kida et al., 2010), though direct comparison is difficult due to the subdivision by environmental regime done here. The GPM V05 passive algorithm (i.e. GPROF) employs the above method to improve detection of light rain below the sensitivity limits of DPR (GPM Science Team, 2017).

### 3.2 Profiles of hydrometeors

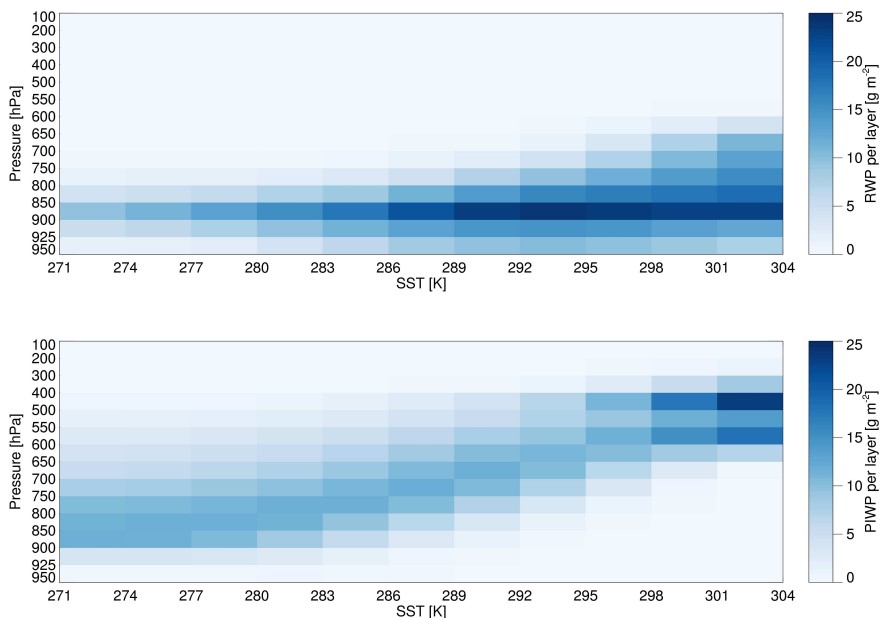

**Figure 2.** First principal components of RWC and PIWC from CloudSat for warm rain scenes, shown as path quantities per layer, RWP and PIWP respectively, to match the units expected by CRTM.





Profiles of hydrometeor species are required to run a realistic radiative transfer (RT) scheme as part of the forward model. Further, the surface rain rate depends on the rainwater content in the lowest atmospheric level, not a column total. Vertical information is however effectively nonexistent in the $T_B$ vector, as the emissivity of drops is not strongly tied to temperature or pressure. Global model data are insufficient to aid in vertical constraints due to the spatiotemporal heterogeneity of clouds and

precipitation. Instead, principal component (PC) analysis can reduce the dimensionality of the problem, simplifying treatment of hydrometeor profiles in the retrieval.

Two years of data from the CloudSat 2C-Rain-Profile product (Lebsock and L'Ecuyer, 2011) were analysed to determine the principal components that best describe hydrometeor profile variability for warm rain. These are separated by SST and lightly smoothed, with the first PC of rain water content (RWC) and precipitating ice water content (PIWC) shown in Fig. 2. The first

PCs of RWC and PIWC describe 63% and 51% of the total variability, respectively. Covariances between the PCs of RWC and PIWC are also calculated and included in the a priori covariance matrix for raining scenes.

Attempting to retrieve more than one PC of each species is unproductive and can lead to non-convergent retrievals. The second PC of each species is effectively a vertical redistribution of the first PC in altitude, i.e. more RWC near the surface and less RWC higher up or vice versa. Since the $T_B$ vector is, to first order, sensitive to total columnar liquid, inclusion of more

PCs is not useful for a passive retrieval, a topic explored further in Sect. 6.

### 3.3 Drop size distributions

For this study, the normalized gamma distribution is used to characterize raindrop distributions (Testud et al., 2001). This functional form, given below as the number concentration of drops as a function of drop diameter, $N(D)$, approximates DSDs found in nature with fidelity (Bringi et al., 2003) though not perfectly (Thurai et al., 2017). The normalized gamma distribution

allows comparison of DSDs with different rain rates and water contents due to the normalized intercept parameter ($N_w$). The median volume diameter ($D_0$) is related to the mass-weighted mean diameter ($D_m$) via the shape parameter ($\mu$), and $\Gamma$ is the gamma function.

$$N(D) = N_w f(\mu)(\frac{D}{D_m})^\mu e^{-(\mu+4)D/D_m} \qquad \text{where} \ \ f(\mu) = \frac{6}{4^4}\frac{(\mu+4)^{\mu+4}}{\Gamma(\mu+4)}, \ \ \frac{D_o}{D_m} = \frac{\mu+3.67}{\mu+4}, \ \ N_w = \frac{3.67^4 RWC}{\pi \rho_w D_o^4} \qquad (1)$$

In situ disdrometer measurements from GPM Ground Validation field campaigns are used to quantify the error in forward

modeled $T_B$s given a range of DSDs. These observations are split into extratropical and tropical locations. The extratropical sites are near Seattle and Helsinki, from the OLYMPEX (Houze Jr et al., 2017) and Light Precipitation Validation Experiment campaigns, respectively. The tropical observations are from Gan Island, Manus Island, and Darwin, Australia. All these sites are an oceanic subset of those used by Dolan et al. (in press), providing the parameters that describe a modified gamma distribution along with liquid water content. Following Dolan et al. (in press), PC analysis of the disdrometer data reveals leading modes of

variability in the DSD parameters that suggest convective and stratiform regimes of rainfall, coloured in Fig. 3. Representative values of these parameters will be used in the retrieval and are separated into these regimes and by location, i.e. tropical or extratropical. For the extratropics, the assumed DSD parameters are $\mu$=9 and $D_0$=0.75 mm for the stratiform case, and $\mu$=-1





and $D_0$=1.8 mm for the convective case; they are $\mu$=7 and $D_0$=0.83 mm, $\mu$=0.5 and $D_0$=1.6 mm for tropical stratiform and convective cases, respectively.

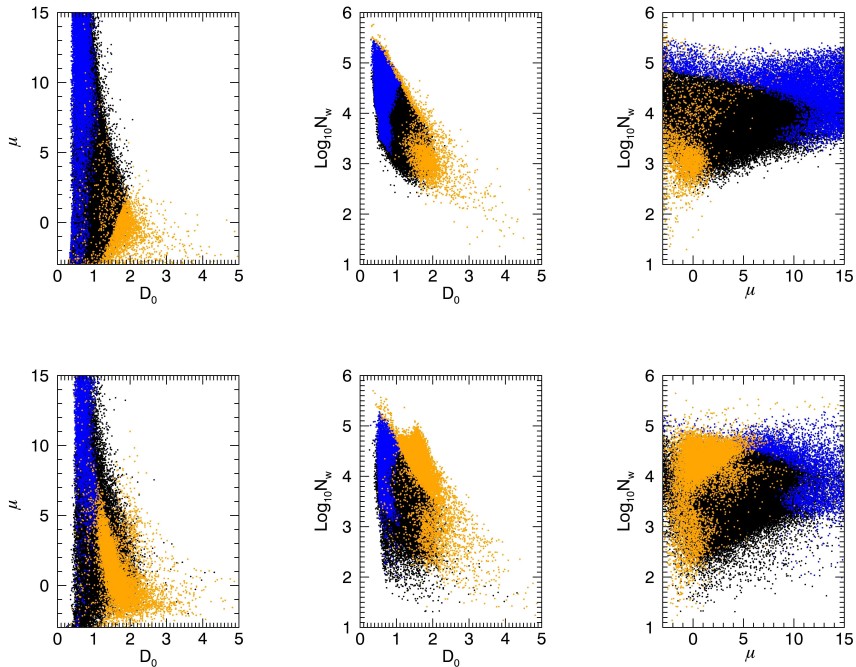

**Figure 3.** Disdrometer data from extratropical (top row) and tropical (bottom row) ocean sites. Blue and orange points were determined via PC analysis to be analogous to stratiform and convective DSDs, respectively, whereas black points did not fall into those categories.

To test the DSD variability's effect on radiances, the disdrometer data were used in a simple model with Eddington absorption (Kummerow, 1993) and Mie scattering modules. The RT model was run with a prescribed atmosphere and surface state, with a 150 g m$^{-2}$ liquid cloud from 925 to 850 hPa. GMI frequencies and viewing geometry are assumed. Rainwater exists below the cloud base, with the RWC values coming from the disdrometer data and distributed evenly. As seen next, the RT model diagnoses different radiometric characteristics of the stratiform and convective DSDs, leading the analysis here and the retrieval described later to delineate between the two.

Figure 4 shows the correlation between $T_B$s at GMI frequencies and three of the DSD parameters ($\mu$, $D_0$, $N_w$) as well as RWC and rain rate, broken up into the stratiform and convective regimes from the high latitude data from Fig. 3, using the simple model described above. The strong positive correlations between low frequency $T_B$s and RWC reveal why it makes more sense to retrieve RWC than any of the DSD parameters, which exhibit weaker correlations that are more channel dependent. As radiances correlate most strongly with rainwater content and weakly with parameters representing the rain's microphysical properties, the spectrum of DSD variability requires simplification to reduce the inverse problem's dimensionality. This binary





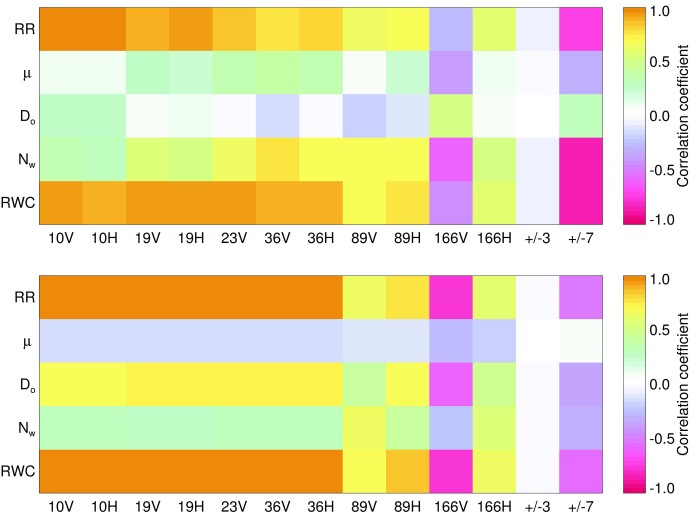

**Figure 4.** Correlations of $T_B$ at GMI frequencies with the DSD parameters (Eq. 1) as well as rain rate and RWC, as derived from disdrometer measurements run through a RT model. Convective DSDs (top) and stratiform DSDs (bottom) are shown for extratropical ocean cases.

**Table 1.** Effects on $T_B$ at top of atmosphere from cloud water and rainwater alone. Surface conditions are SST=281 K and wind=5 m s$^{-1}$, with water vapour and temperature profiles representative of such an ocean scene. Liquid water path is 100 g m$^{-2}$ for both rain and cloud water. In both cases the hydrometeors reside between 925 to 975 hPa. GMI's 183 GHz channels are not included due to the invariance of water vapour here. Radiometric signals from rainwater are separated into emission (emis.) and scattering (scat.) as described in the text, with the net effect also given. All units are $\Delta K$ except for the top row, which is in K.

| | 10V | 10H | 19V | 19H | 23V | 37V | 37H | 89V | 89H | 166V | 166H |
|---|---|---|---|---|---|---|---|---|---|---|---|
| Clear sky [K] | 160.16 | 82.80 | 178.53 | 104.79 | 199.25 | 204.04 | 133.86 | 243.35 | 192.73 | 269.36 | 261.03 |
| Cloud | +0.40 | +0.65 | +1.02 | +1.77 | +1.29 | +2.73 | +5.29 | +5.28 | +13.1 | +1.40 | +3.64 |
| Strat. rain (net) | +0.58 | +0.96 | +1.73 | +3.02 | +2.27 | +5.02 | +10.37 | +5.58 | +26.11 | +0.12 | +4.72 |
| Conv. rain (net) | +2.10 | +3.61 | +4.10 | +8.30 | +4.17 | +5.33 | +16.4 | +2.18 | +16.9 | +0.29 | +2.83 |
| Strat. rain (emis.) | +0.59 | +0.97 | +1.80 | +3.11 | +2.42 | +5.90 | +11.5 | +15.2 | +37.6 | +3.10 | +8.16 |
| Conv. rain (emis.) | +2.44 | +4.02 | +6.34 | +10.9 | +7.70 | +13.7 | +26.5 | +11.7 | +29.0 | +1.83 | +4.79 |
| Strat. rain (scat.) | -0.01 | -0.01 | -0.07 | -0.09 | -0.15 | -0.88 | -1.08 | -9.59 | -11.54 | -2.98 | -3.44 |
| Conv. rain (scat.) | -0.34 | -0.41 | -2.24 | -2.68 | -3.53 | -8.34 | -10.2 | -9.51 | -12.1 | -1.54 | -1.96 |

classification is a way to simplify the problem without treating all DSDs as the same, in line with there being limited signal to solve for the DSD but some information related to the DSD existing in the $T_B$s.

To view the competing radiance signals more quantitatively, the two DSD regimes' impacts on $T_B$ are enumerated via a simple model in Table 1. Nearly identical to the model setup used above, here we first run the clear sky case, then with




$100\,\mathrm{g\,m^{-2}}$ liquid cloud, then simulate a $100\,\mathrm{g\,m^{-2}}$ rain cloud. The rain cloud has a fixed RWC but the DSD varies as per the regimes defined above for the extratropical case. To pull apart the signals, no cloud water was included, and the model was run once with rainwater emission artificially set to zero and scattering turned off in another run. Notable are the similar signals between cloud alone and stratiform rain, and the strong channel dependence of the signals from rainwater.

5      In an attempt to circumvent the issue of DSD variability while accounting for the inherent forward model uncertainty of assuming a DSD, these errors are quantified in a way intended to reduce the dimensionality of the problem without ignoring it. This stems from the $T_B$ vector containing information on the DSD, but not enough to be solved for explicitly. The forward model parameter error, given below as the variance ($\sigma^2$) per frequency ($\nu$) stemming from an assumed drop distribution (e.g. convective, $DSD_{conv}$) is defined as:

10    $$\sigma^2_{conv}(\nu) = var(T_B(\nu, DSD_{conv}) - T_B(\nu, DSD_{actual})) \tag{2}$$

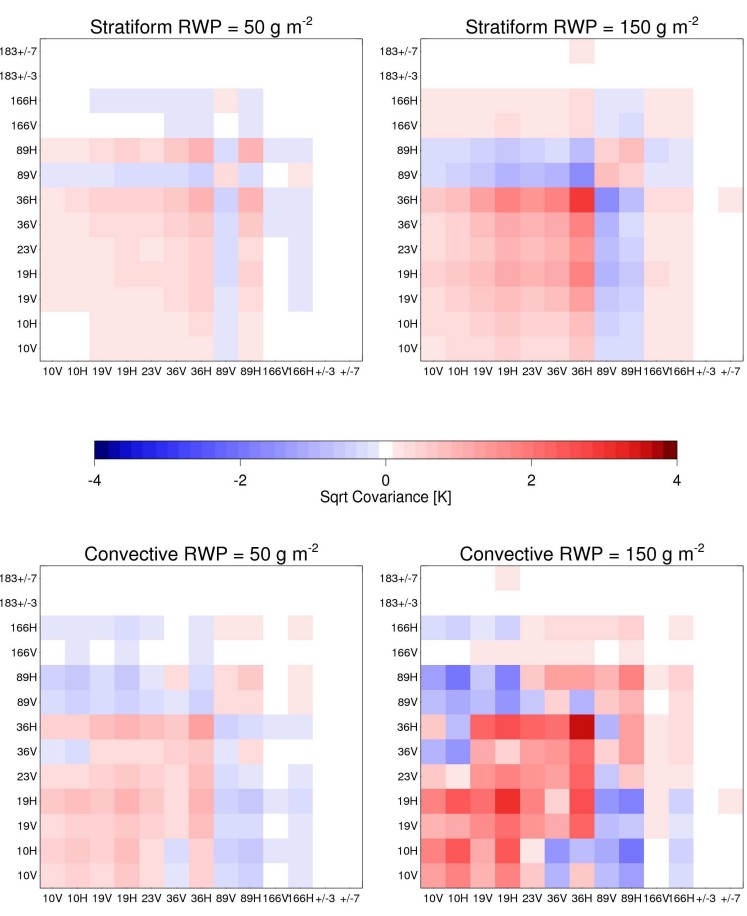

**Figure 5.** Error covariances due to DSD variability observed at extratropical ocean sites. The RWP values of 50 and $150\,\mathrm{g\,m^{-2}}$ are nominal. Covariances are in units of K, with negative covariances given as $-\sqrt{|S_y|}$, to aid interpretation.





Figure 5 translates the simple model containing in situ DSD data into error covariance matrices usable by the retrieval, via Eq. 2. Shown are error covariance matrices calculated for both stratiform and convective DSD observations at the extratropical sites for two nominal rain water path (RWP) values. These values are in line with DSDs connected to collision-coalescence processes (Dolan et al., in press) and thus appropriate for warm rain. To apply these analyses of in situ data as realistically as

possible, the errors and DSD assumptions derived from extratropical and tropical sites are dependent on the observed latitude.

The result of this analysis is an estimate of forward model error at GMI frequencies caused by the assumption of a DSD for rain in each regime. Since this analysis used the observed variability of the DSD parameters for given RWC values, the resultant error covariance matrices can be scaled as a function of RWC in the retrieval without further assumptions. The inclusion of covariances between channels' errors (i.e. off-diagonal matrix elements) is key, as many of the errors caused by assuming a

DSD are highly correlated between nearby channels.

## 4   Retrieval description

The following subsections detail how the retrieval algorithm treats non-raining, drizzling, and warm raining pixels. Its progression through these outcomes is described via flowchart in Fig. 6. The non-raining retrieval is always run first, with either non-convergence or high retrieved LWP signalling the need for the warm rain retrieval to be run. Drizzle is effectively an

in-between case, where the non-raining forward model is sufficient to match the observed $T_B$ vector but the retrieved LWP exceeds the drizzle onset threshold (Fig. 1). All 13 GMI channels are used in every case.

### 4.1   Non-raining algorithm

The CSU 1DVAR (Duncan and Kummerow, 2016) was originally developed as a non-scattering retrieval for the so-called "ocean suite" parameters over water: total precipitable water, $10\,\mathrm{m}$ wind speed, cloud LWP, and SST. It is a variational (optimal

estimation) algorithm that iterates to find an optimal geophysical state that best matches the observed $T_B$ vector within the bounds of a priori knowledge of the geophysical state (Rodgers, 2000). This is done via a physical forward model tailored to the radiometric sensitivities of the variables being retrieved, using Gauss-Newton iteration. Mathematically, the iterative process endeavours to find a state vector ($x$) that minimizes a cost function ($\Phi$) and yields a metric of fit ($\chi^2$) to the observed radiances:

$$\Phi = (y - f(x,b))^T S_y^{-1} (y - f(x,b)) + (x - x_a)^T S_a^{-1} (x - x_a), \qquad \chi^2 = (y - f(x,b))^T S_y^{-1} (y - f(x,b))/N_{chan} \qquad (3)$$

Here $y$ is the observation vector, $f$ is the forward model, $b$ contains all non-retrieved elements of the forward model, $x_a$ is the a priori state vector, and $S_a$ and $S_y$ represent the error covariance matrices of the a priori and observation vectors, respectively. $S_y$ for the non-raining retrieval is the same as that given by Duncan and Kummerow (2016). The cost function balances knowledge of the prior state with confidence in the observations to find an optimal retrieved state. The fit metric ($\chi^2$) is normalized by the

number of channels used.





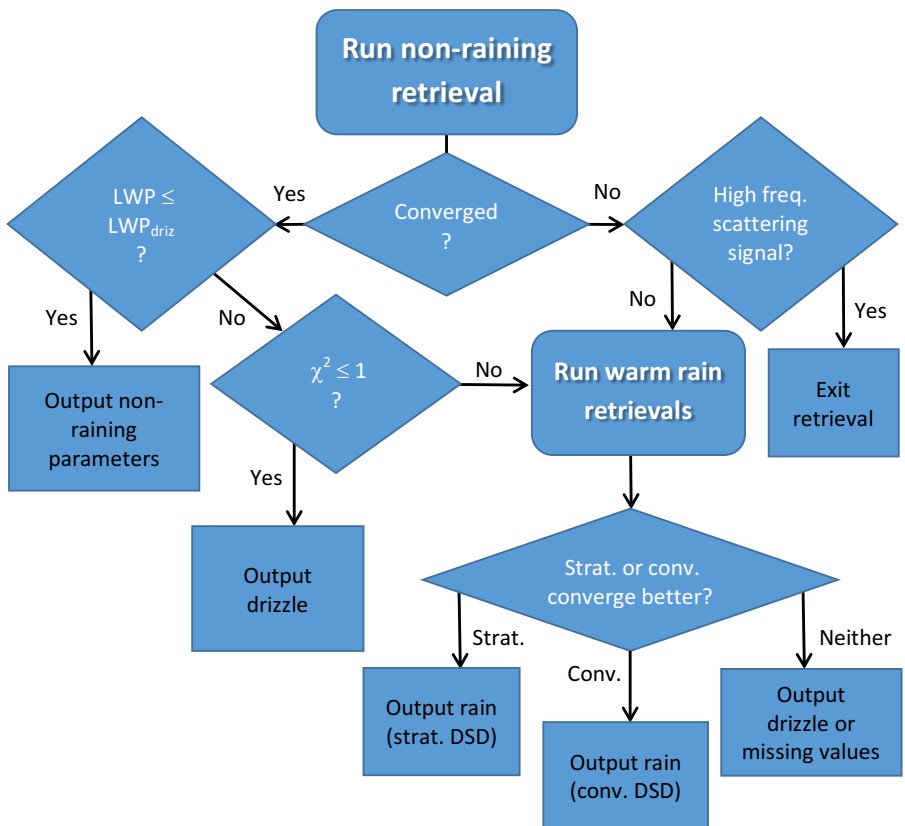

**Figure 6.** Algorithm flowchart for the 1DVAR for warm rain.

The non-raining CSU 1DVAR solves for six parameters: wind speed, liquid water path, SST, and coefficients of the first three PCs of water vapour. Just as described in Sect. 3.2, PCs reduce the dimensionality of the water vapour profile. To make the problem more Gaussian, LWP is retrieved in logarithm space but with effectively no constraint by the prior. The a priori states for SST, wind, and water vapour come from a global model, as do sea level pressure and the temperature profile. For this

5  study, the GEOS5 FP-IT model (Lucchesi, 2013) was used.

The forward model for the CSU 1DVAR uses the Community Radiative Transfer Model (CRTM) v2.3.3 coupled with the FASTEM6 emissivity model (Liu and Weng, 2013; Kazumori and English, 2015). There are 16 vertical layers from the surface up to $100\,\mathrm{hPa}$. Cloud liquid water is evenly distributed from 925 to $850\,\mathrm{hPa}$ with a cloud drop effective radius of $12\,\mu\mathrm{m}$, a value consistent with observations (Lebsock et al., 2008).

10  ## 4.2  Drizzle

Drizzle is poorly characterized by passive measurements alone, and so the drizzle retrieval depends heavily on CloudSat data. Conditions in which the non-raining (non-scattering) retrieval converges with a high quality of fit ($\chi^2 < 1$) are not necessarily





non-raining for the reasons mentioned in Sect. 3.1. Thus, if retrieved LWP is greater than the CloudSat-derived drizzle onset threshold (Fig. 1), LWP is partitioned into cloud and rain water. Not all extra water is partitioned into drizzle, with some of the extra water remaining as cloud water as discussed by Wentz and Spencer (1998). RWP is defined thus, with $LWP_{driz}$ determined from Fig. 1 using the a priori SST and TPW states:

$$RWP = \delta LWP(1 - \frac{1}{\sqrt{\delta LWP}}), \text{ where } \delta LWP = LWP - LWP_{driz}(SST, TPW) \tag{4}$$

The resultant drizzle rate is a function of RWP. Because no information exists on the drops' distribution or altitude, a simple regression relationship derived from the 2C-Rain-Profile dataset is used to calculate a rain rate, linearly related to RWP and subset by SST regime. In most regimes the relationship is on order of $70\,\mathrm{g\,m^{-2}}$ of RWP per $1\,\mathrm{mm\,h^{-1}}$ of rain rate. Relative to the CloudSat estimate, this regression relationship tends to underestimate heavy rain rates and slightly overestimate light rain rates.

The predominantly non-scattering scenario with drizzle is quite common, especially at high latitudes, and forms a plurality of global scenes with retrieved rain. Shallow clouds with high liquid water contents often converge well in the non-scattering retrieval if there is a lack of significant snow or mixed phase hydrometeors. The high frequency channels on GMI—166 GHz and higher—are sensitive to scattering from frozen hydrometeors (Gong and Wu, 2017). Because this retrieval is for warm rain only, a significant departure between observed and simulated $T_B$ at 166V, 166H, and 183±7V is a sign that the warm rain retrieval should not be run because the forward model is inadequate (Fig. 6). In the algorithm, this condition is met if the mean observed minus simulated $T_B$ of those three channels is less than $-8\,\mathrm{K}$, in which case missing values are output.

## 4.3 Warm rain retrieval

For cases where the non-scattering retrieval fails, or cases in which $LWP > LWP_{driz}$ but $\chi^2 > 1.0$, indicating a fit to the $T_B$s that exceeds assumed errors, the warm rain retrievals are run subsequent to the non-scattering retrieval. The number of retrieved parameters drops from six to four: PC1 of RWC, PC1 of PIWC, LWP, and PC1 of water vapour. This is necessary due to the limited information content afforded by the $T_B$ vector in raining conditions, where sensitivity to the surface and water vapour are superseded by signals from hydrometeors. The a priori wind and SST are thus held constant; attempting to retrieve wind speed or SST tends to degrade retrieval of the other parameters. Even with four variables, the a priori errors on LWP and PC1 of water vapour are decreased so as to discourage unphysical behaviour in the retrieval. Raining scenes can exhibit 1.5-3.5 degrees of freedom for signal (DFS) given these four retrieved parameters, indicating that even with four variables the problem is information-limited.

A key element of the rain retrieval is its dynamic observation error covariance matrix. In theory, $S_y$ should contain all the uncertainties of the forward model, forward model parameters, and instrument noise. In practice, this means adding the non-scattering retrieval's errors with the errors for a given RWP. As described in Sect. 3.3, the forward model error caused by assuming a DSD is a function of RWP. Dynamic adjustment of observation errors based on the retrieved scene's characteristics is not commonly done in either retrievals or DA schemes; an analogue is Lean et al. (2017), which uses a proxy for cloud amount to determine errors, a scheme akin to a dynamic error assignment though not specific to DSD assumptions. Interestingly, the





**Table 2.** GMI channel error variances used during iteration for randomly chosen pixels from the scene in Fig. 7. The square roots of error variances are given so as to be in K, and RWP is given in $\mathrm{g\,m^{-2}}$.

| DSD | RWP | 10V | 10H | 19V | 19H | 23V | 36V | 36H | 89V | 89H | 166V | 166H | 183±3 | 183±7 |
|-----|-----|-----|-----|-----|-----|-----|-----|-----|-----|-----|------|------|-------|-------|
| - | 0 | 1.51 | 1.13 | 1.86 | 2.43 | 2.60 | 1.43 | 2.32 | 1.61 | 3.42 | 1.83 | 2.71 | 5.61 | 3.22 |
| Stra. | 18 | 1.52 | 1.14 | 1.87 | 2.44 | 2.61 | 1.45 | 2.35 | 1.63 | 3.47 | 1.84 | 2.72 | 5.61 | 3.23 |
| Stra. | 309 | 1.54 | 1.19 | 2.02 | 2.81 | 2.76 | 2.07 | 4.23 | 2.03 | 3.51 | 1.98 | 2.80 | 5.61 | 3.24 |
| Conv. | 79 | 1.63 | 1.64 | 2.02 | 2.93 | 2.71 | 1.63 | 3.21 | 1.69 | 3.62 | 1.85 | 2.73 | 5.61 | 3.24 |
| Conv. | 195 | 2.26 | 3.43 | 2.86 | 4.91 | 3.24 | 2.37 | 5.27 | 1.97 | 3.91 | 1.98 | 2.80 | 5.61 | 3.24 |

largest errors given by Lean et al. (2017) are at the 19H and 37H channels for GMI, in line with the results of Fig. 5 for large RWP.

The vertical distribution of RWC is also assumed by virtue of using only one PC of RWC. This too affects forward model errors, and was quantified by similar analysis of CloudSat retrievals, also as a function of RWP. These values are added to the
$S_{y,rain}(RWP)$ depicted in Fig. 5. This particular error source has little impact on the retrieval as channel errors are effectively zero for most channels, maximizing at $3.5\,\mathrm{K^2}$ for high RWP at 36H in the convective case. Because the errors add in quadrature, these are mostly insignificant.

Summing $S_{y,non-scat} + S_{y,rain}(RWP)$ yields the observation error covariance matrix used in the iteration (though some care needs to be taken to ensure that it remains positive definite). Because RWP is retrieved, the matrix is updated with
every iteration. This complicates the iteration process, but it is based in the physics of the situation—heavier rainfall begets larger uncertainties. Examples of observation error channel variances are given in Table 2 for randomly selected RWP values from an extratropical case with both (i.e. convective and stratiform) DSD assumptions. Note that the DSD assumptions and corresponding errors depend on latitude—retrievals within the tropics (30°N to 30°S) use a different DSD from those in the extra tropics, as described in Sect. 3.3.

The stratiform and convective rain retrievals are run side by side. Whichever converges with a better fit to observations (lower $\chi^2$) is output. If neither converges, the output is either that from the non-scattering retrieval, i.e. non-scattering drizzle, or missing values (see Fig. 6). The convective case is treated the same as the stratiform case—only the DSD parameters and observation errors differ. For both cases, the resultant rain rate is averaged from the three lowest altitude layers of RWC in the forward model. This includes the standard assumption that drops reach their terminal fall speed. No explicit evaporation model
is included due to the lack of true vertical information.

The forward model for warm rain builds upon the non-raining forward model but requires some modification, as CRTM does not currently support functional variations in DSD. Thus, the warm rain forward model uses both CRTM and the Eddington absorption model (Kummerow, 1993) with Mie code modules. The Eddington codes are the same codes used for the GPROF a priori database creation and the RT simulations described in Sect. 3.3. In practice, this means calling CRTM and then running
Eddington twice—once with the RWC and PIWC profiles included and once without—then differencing the two and adding



this to the CRTM-derived $T_B$ vector. This avoids forward model discontinuity between raining and non-raining pixels, but is not ideal and computationally expensive.

## 5   Proof of concept

### 5.1   Case studies with space-borne radars

CloudSat's sensitivity to light rain rates makes it a useful point of comparison, although the orbits of GPM and CloudSat result in limited high quality matchups. This section includes one case with GMI, DPR, and CloudSat observations in the North Atlantic, and one case with GMI and CloudSat off the coast of France.

Figure 7 compares the CSU 1DVAR, GPROF, DPR, and CloudSat rain rates for a coincident overpass in the North Atlantic on June 1st, 2015. The figure's projection orients the CloudSat ribbon horizontally, with CloudSat reflectivities shown at the

top of the figure. GPROF and DPR underestimate rainfall relative to CloudSat whereas the CSU 1DVAR estimates are closer in magnitude to CloudSat, as seen in the line plot within Fig. 7. DPR misses the majority of the raining pixels seen by CloudSat, as the reflectivites are generally below DPR's detection threshold. From 49°N to 51.5°N, the region of overlap for the three sensors, the CloudSat 2C-Rain-Profile product has a mean rain rate of $1.30\,\mathrm{mm\,h^{-1}}$ whereas GPROF and DPR NS measure 0.58 and $0.13\,\mathrm{mm\,h^{-1}}$, respectively. The CSU 1DVAR mean for the same pixels is $1.87\,\mathrm{mm\,h^{-1}}$, though a few pixels failed

to converge. This is an encouraging result, showing that warm rain from the variational algorithm is of the same order as that from CloudSat.

The freezing level is denoted by a grey line in the top panel of Fig. 7, calculated from ancillary data. This lies above most of the cloud tops seen by CloudSat, indicating that most of the clouds are probably liquid. The CSU 1DVAR converges for many of these pixels, excepting a few near 52°N and 50°N, where CloudSat shows stronger convection and radar echoes above the

freezing level. The GPM and CloudSat overpasses were $10\,\mathrm{min}$ apart, which may explain some incongruity in the pixels that converged, especially with regard to convective clouds.

Figure 8 provides a closer look of a raining system in the Atlantic, a scene from March 30th, 2016 off the coast of France. In this figure, CloudSat reflectivities show a complex scene with multiple cloud layers and cloud depths ranging from $1\,\mathrm{km}$ to $8\,\mathrm{km}$. The second panel holds retrieval results from 2C-Rain-Profile, colour-coded to differentiate between liquid and ice

portions of the cloud. CloudSat shows significant rainwater content near the surface that translates into rain rates of about $4\,\mathrm{mm\,h^{-1}}$. This is in contrast to the GPROF rain rates, which are all less than $0.5\,\mathrm{mm\,h^{-1}}$. As with the previous case, this is not surprising because GPROF's a priori database is based upon DPR and most of the CloudSat reflectivities seen from 46°N to 47°N in Fig. 8 are below the sensitivity limit of DPR. This raining system is on the edge of the GMI swath, so no DPR data are available.

The CSU 1DVAR mostly performs well in this scene. On the right of the figure where the clouds are shallow and mostly liquid, it retrieves rain rates on the order of CloudSat and much higher than GPROF. As the cloud deepens and non-liquid hydrometeors dominate, it fails to converge—the forward model is insufficient due to the transition away from warm rain. In fact, the apparent overestimation of rain rates on the right side of the figure may be due to CloudSat missing some rainwater;





GMI senses total columnar liquid, whereas CloudSat is mostly blind in the lowest kilometer of the atmosphere and thus may miss rainwater near the surface (Liu et al., 2016).

On the northern edge of the retrieved rain band in Fig. 8 exists a transition zone with low retrieved rain rates in an area with moderate CloudSat rainfall. This violates the assumptions of the forward model, but not strongly enough to cause non-

convergence. Instead, the scattering signal of mixed phase hydrometeors appears to cancel out the rain drops' emission signal, and the algorithm reaches convergence with limited rainfall, albeit with a fairly poor fit. As with the previous case, about $9\,\mathrm{min}$ elapsed between the overpasses, so the characteristics of the clouds and precipitation may have evolved. The plane parallel forward model could also be a cause of discrepancies at the rain band's edge.

## 5.2   Case studies with ground radar

In this section two GPM overpasses of the PAIH ground radar are examined. Due to GPM's orbit and the radar's location south of Alaska, it is an ideal location for comparisons between high latitude oceanic GPM observations and a polarimetric ground radar. For this analysis, the focus is on precipitation away from the coastline, as emission from nearby land is a contaminating factor in precipitation retrievals; indeed the CSU 1DVAR does not run if a pixel contains land contamination.

The first case, shown in Fig. 9, is from an overpass on July 12th, 2015 with scattered showers visible from PAIH. DPR does

a fairly good job of seeing these showers, although it misses some of the lightest raining pixels observed by PAIH. GPROF picks up the strongest region of rainfall but underestimates the rain rate relative to PAIH and misses the weaker showers. This scene proves challenging for the CSU 1DVAR as well. This region is covered with retrieved liquid cloud, including some pixels above the drizzle onset threshold that fit the forward model well. Contrasting these pixels with PAIH, some are not raining to the surface while others are below the drizzle threshold but do indeed seem to be raining. The few pixels raining

hardest according to PAIH, DPR, and GPROF do not converge in the iteration, in line with significant mixed phase or frozen hydrometeors present and echo top heights of $3\,\mathrm{km}$ to $5\,\mathrm{km}$ observed by DPR. So while this scene is nearly ideal for the CSU 1DVAR rain retrieval, in that it rarely violates the assumptions of the forward model, the assumption of a drizzle onset threshold proves too simplistic to accurately capture drizzling versus non-drizzling liquid clouds in this scene.

Figure 10 shows a second ground radar matchup with GPM, from June 29th, 2015. A stronger band of rain is identified

consistently by DPR and GPROF, and they agree on the general magnitude of precipitation, but PAIH is slightly higher. The CSU 1DVAR gets the right general shape of this rain band but mostly overestimates the rain rates compared to the other estimates. Examination of the fit metric ($\chi^2$) shows that much of this band exhibited relatively poor fits to the observations.

Further analysis of the DPR and PAIH data in Fig. 10 indicates that the forward model assumptions were violated for many of these raining pixels (not shown). DPR retrieved echo top heights of $1.5\,\mathrm{km}$ to $4.0\,\mathrm{km}$, with bright bands evident in most pixels

between $1.6\,\mathrm{km}$ and $1.8\,\mathrm{km}$. The existence of these bright bands picked up by DPR demonstrates that there were significant areas of mixed phase hydrometeors present, something absent from the forward model. Most of the raining pixels in the figure reached convergence with the convective DSD assumptions but many still exhibit relatively poor fits to the observations. This points to the utility of $\chi^2$ as a marker of trustworthiness for retrieved parameters (Elsaesser and Kummerow, 2008), suggesting caution in interpreting such pixels that display errors larger than those assumed.



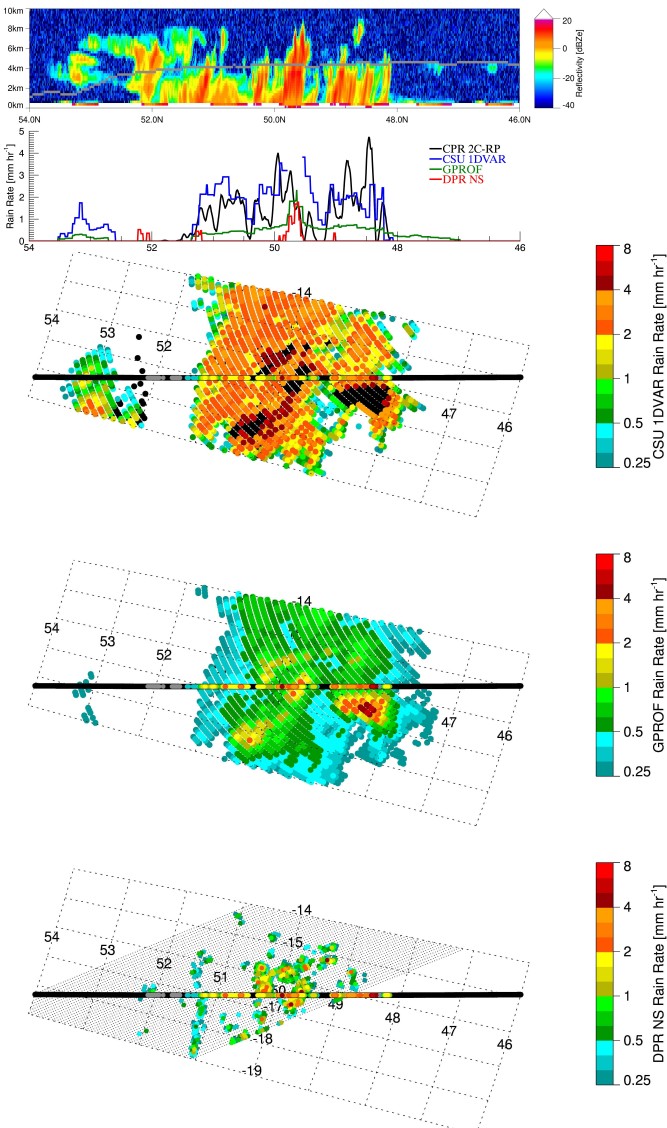

**Figure 7.** GPM and CloudSat rain rates for coincident overpasses in the North Atlantic on June 1st, 2015. The top panel shows CloudSat reflectivities with a grey line indicating the freezing level. The second panel gives rainfall rates along the CloudSat track. For the bottom panels, black along the CloudSat track indicates no rain and grey is snow or mixed phase precipitation. Black pixels for the 1DVAR signify non-convergence. In the final panel, black stippling marks the extent of the DPR NS swath.

## 5.3 Statistical analysis against DPR

Moving beyond case studies, twelve months of data from the 1DVAR retrieval were compared against DPR rain estimates to assess the representativeness of the analysed cases. Only pixels within the DPR matched scan (MS), containing both Ku- and





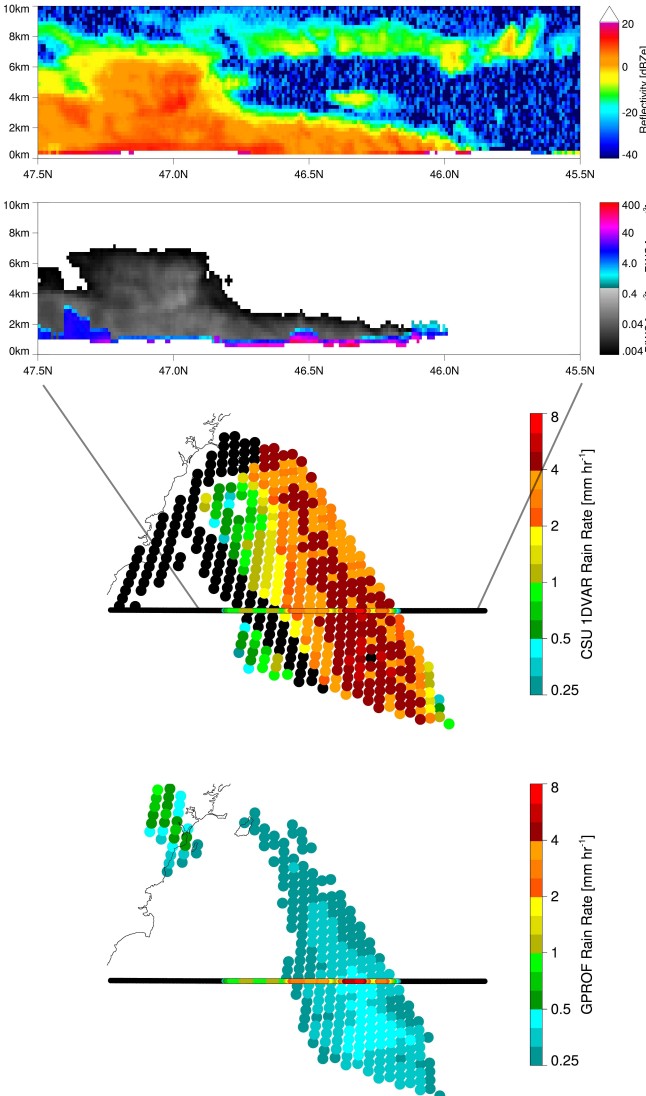

**Figure 8.** Shallow rain and mixed phase cloud off the coast of France, March 30th 2016. The top panel is CloudSat reflectivities while the second panel shows 2C-Rain-Profile RWC and PIWC profile retrievals from the same scene. The bottom two panels contrast CloudSat rain rates with those of the CSU 1DVAR and GPROF. Colour conventions follow those of Fig. 7.

Ka-band observations, were considered. DPR pixels were averaged into the GMI 23 GHz FOV via the same spatial weighting scheme used to create the GPROF database. The matched data constitute over 120 million coincident observations spanning September 2014 through August 2015, 20 million of which contain positive rain rates in one or both datasets. Here a threshold of $0.2\,\mathrm{mm\,h^{-1}}$ defines positive rain to avoid the distribution's tail that arises from averaging.



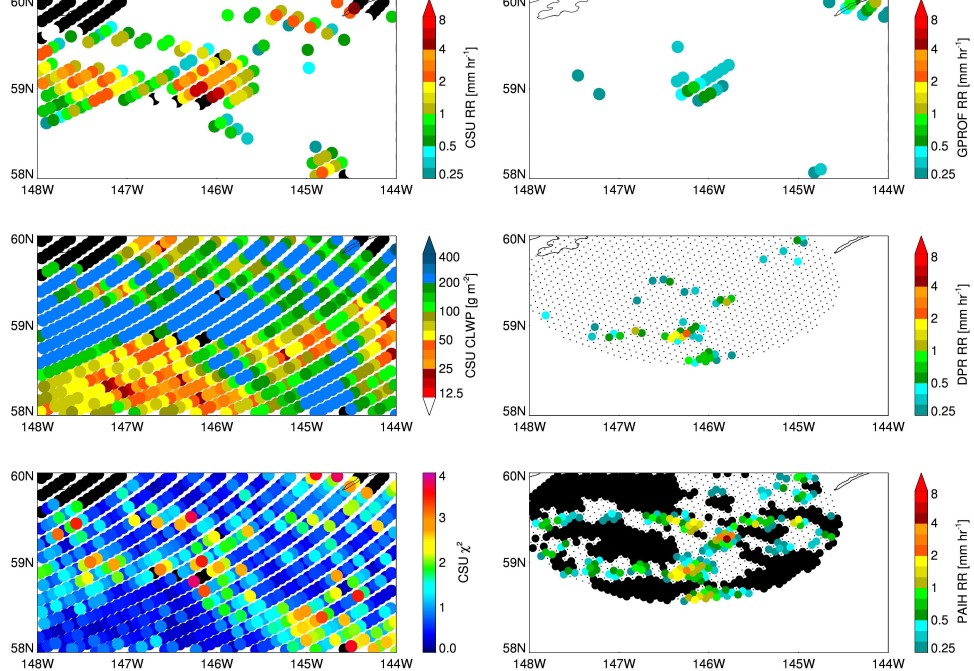

**Figure 9.** Middleton Island (PAIH) radar compared with GPM products and CSU 1DVAR retrievals from July 12th, 2015. The left column contains CSU 1DVAR retrievals of rain rate and cloud liquid water path, and quality of fit ($\chi^2$). The right column contains rain rates from GPROF, DPR, and the ground radar. Colour conventions follow those of Fig. 7.

The 1DVAR retrieves rainfall in a significant percentage of pixels where DPR sees no precipitation. Of all pixels where the 1DVAR retrieved rain rates greater than $0.2\,\mathrm{mm\,h^{-1}}$, DPR saw zero rain in 44% of them, with an overall mean rain rate of $0.24\,\mathrm{mm\,h^{-1}}$ versus $1.43\,\mathrm{mm\,h^{-1}}$ from the 1DVAR. This discrepancy is biggest for the drizzle retrievals, where DPR retrieves zero rain rates for 59% of GMI pixels found to be drizzling. However, of all these cases with zero DPR rain and positive
5  rain from the 1DVAR, 80% are below $2\,\mathrm{mm\,h^{-1}}$. This indicates that it is almost always light rain that the 1DVAR picks up, consistent with the sensitivity limitations of DPR. In the opposite view, the 1DVAR misses a relatively small percentage of definite raining cases from DPR and effectively none at higher rain rates. The 1DVAR ascribes non-raining to only 2.3% of DPR retrievals greater than $0.5\,\mathrm{mm\,h^{-1}}$ and a mere 0.03% of DPR retrievals greater than $2\,\mathrm{mm\,h^{-1}}$. This result speaks to the sensitivity of the 1DVAR and its forward model, consistent with Duncan and Kummerow (2016).
10  Additional analysis elucidates some physical causes for 1DVAR versus DPR discrepancies beyond those of the sensors' differing sensitivities. For example, pixels where the 1DVAR fails to converge are more often characterized by the presence of a detectable bright band and higher DPR-detected echo top heights. This is most stark for pixels screened out due to high frequency scattering, which exhibit bright bands in 42% of their area on average and have echo top heights over double those of 1DVAR-retrieved drizzle pixels, 5.3 versus $2.5\,\mathrm{km}$. This fits the hypothesis that most precipitation missed by the 1DVAR





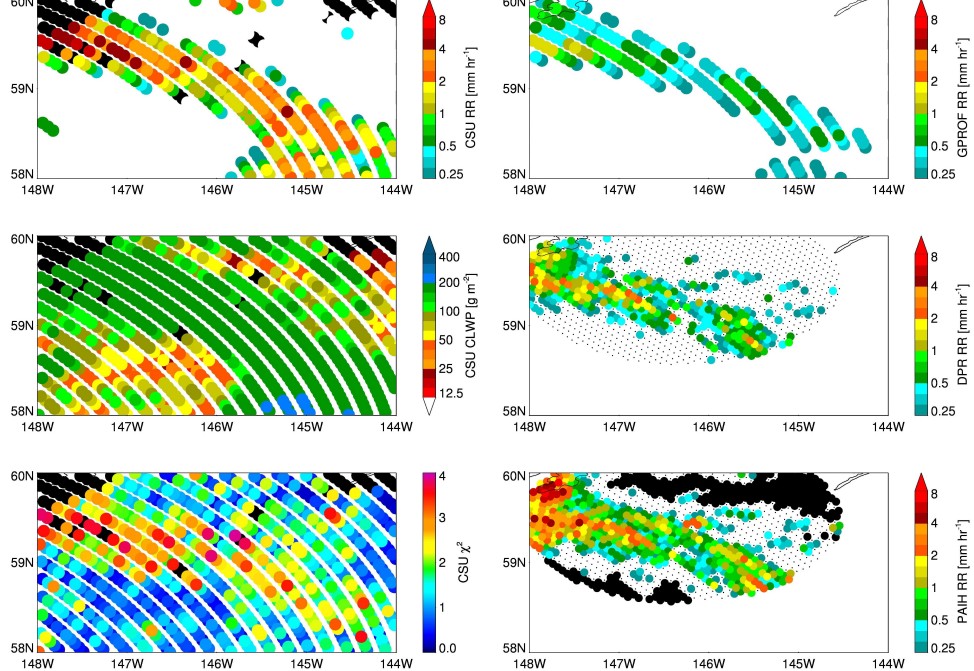

**Figure 10.** Middleton Island (PAIH) radar compared with GPM products and CSU 1DVAR retrievals from June 29th 2015. The left column contains CSU 1DVAR retrievals of rain rate and cloud liquid water path, and quality of fit ($\chi^2$). The right column contains rain rates from GPROF, DPR, and the ground radar. Colour conventions follow those of Fig. 7.

involves significant mixed phase or frozen hydrometeors. In fact, from the subset of pixels where both DPR and 1DVAR retrieved rain, the echo top heights bear out the algorithm's large-scale separations. Mean echo top heights of 2.7 and $3.4\,\mathrm{km}$ were found for converged stratiform and convective retrievals, respectively. The other main cause for discrepancy is sub-pixel FOV heterogeneity. For pixels where DPR and 1DVAR agree that it is raining, DPR observes much less variability in sub-pixel

5  rain rates. These are therefore more aligned with the forward model, which assumes a plane parallel atmosphere. The sub-pixel standard deviation of rain rates from DPR is $3.2\,\mathrm{mm\,h^{-1}}$ for failed 1DVAR retrievals, compared to $0.5\,\mathrm{mm\,h^{-1}}$ for pixels where the 1DVAR reached convergence.

## 6  Discussion

There are advantages and disadvantages to the variational approach when applied to precipitation retrieval. As shown in the

10  comparisons against radar estimates, the retrieval described here compares favourably in some cases and fails to converge in others, sometimes for observations tens of kilometres apart. This is a function of the simple forward model's ability or inability to adequately represent all radiometrically significant constituents associated with oceanic rainfall. However, the simplicity of



the forward model is dictated by the limited information content from the observed $T_B$ vector. This is the fundamental catch-22 of precipitation retrieval with limited information. Additionally, the 1DVAR approach will perform poorly if the relationship between state and observation vectors surpasses moderately non-linear behaviour (Rodgers, 2000), a key upside of Bayesian integration (Kummerow et al., 2015).

Warm rain is difficult to observe with conventional validation sources and is a small fraction of the total precipitation in many regions (Mülmenstädt et al., 2015), making it very challenging to validate. The limited case studies presented indicate that the 1DVAR can outperform GPROF and the DPR in hand-picked situations, at least relative to CloudSat. It is beyond the scope of this study to exhaustively validate the retrieval, as it is experimental and not intended to be operational, meant instead to suggest a possible way forward for future passive rainfall retrievals to reconcile the distribution of precipitation from the

GPM constellation (Skofronick-Jackson et al., 2017) with those of other estimates (Behrangi et al., 2016). With this in mind, the following discussion probes the presented retrieval's limitations, sensitivities, and implications.

## 6.1   Limitations

This study has shown that DSD effects on forward model error can be dealt with, but other impediments such as partitioning liquid water path are perhaps the main cause of errors with respect to radar rainfall estimates. A globally-derived drizzle onset

threshold can cause high and low biases side by side (Figs. 9 and 10), as the $T_B$s cannot necessarily convey information on cloud life cycle stage, microphysics, or environmental regime that will affect whether or not a cloud is raining. Similarly, because GMI lacks profile information, there is no evaporation model, nor a physical model for drizzle rate. These aspects could conceivably be improved by more extensive use of ancillary data.

The simplicity of the forward model—which accounts for no spatial heterogeneity or 3D radiative transfer effects—is cer-

tainly a limitation. Beam-filling is a challenging obstacle for physical retrievals with an explicit forward model, and can cause high biases in retrieved liquid water (Rapp et al., 2009). In the absence of independent sub-FOV observations, cloud fraction parametrizations or $T_B$-based metrics as a proxy for heterogeneity are not ideal or straightforward to apply during iteration, and neither is post-processing of rain rates after running a physical retrieval. This class of errors is not addressed here, and is expected to cause a general high bias in retrieved liquid water and rain rates, consistent with Figs. 7 and 10.

## 6.2   Sensitivities

A few sensitivity experiments were conducted to investigate the retrieval's robustness. Experiments conducted with additional PCs of RWC and PIWC yielded approximately the same DFS as with one PC, demonstrating that retrieval of additional profile parameters is not possible with the information content available. In fact, the algorithm is quite insensitive to the specific shape of the RWC profile employed. A separate experiment using the mean RWC profile from the PC analysis of CloudSat instead

of the first PC yielded almost identical results in the case studies examined (not shown), due to $T_B$s and rain rate being tied strongly to columnar liquid and not its distribution (Fig. 4).

Another possible sensitivity of variational retrievals is their dependency on the a priori state. To test this, the GPROF retrieval was run before the 1DVAR and its columnar rainwater used for the a priori value of RWP. This had a small impact, increasing



the number of raining pixels on average by about 5% but only changing the mean by 2% as the distribution of rain rates was essentially the same. This more sophisticated prior led to greater convergence rates, with convergence for the stratiform and convective cases 7% more likely. The cases studies shown in Figs. 7 and 9 can be compared with these modified a priori cases seen in the supplementary Figs. A1 and A2, respectively.

Also shown in Figs. A1 and A2 are the sensitivity experiments regarding the drizzle onset threshold. The threshold was modified by adding and subtracting $50\,\mathrm{g\,m^{-2}}$ from the drizzle LWP value. This is a large perturbation, but is about $2\sigma$ of typical LWP posterior errors and approximately the difference between non-precipitating and transitional cloud water paths reported by Lebsock et al. (2008). Increasing the drizzle onset threshold caused a decrease in raining pixels by about 30%, while a decrease in the threshold caused an increase in raining pixels by 50%, with the number of points retrieved as drizzle

changing by a factor of two in each direction. This seems quite significant, but perturbations had a smaller impact on overall accumulations, increasing the average rain rate 9% for the lower drizzle onset, and decreasing the rain rate 7% for the higher onset. Because drizzle rates are generally insubstantial, changes to the drizzle onset threshold may have a large impact on the frequency of light rain but not on global accumulations, though the impacts may be substantial in persistently cloudy regions.

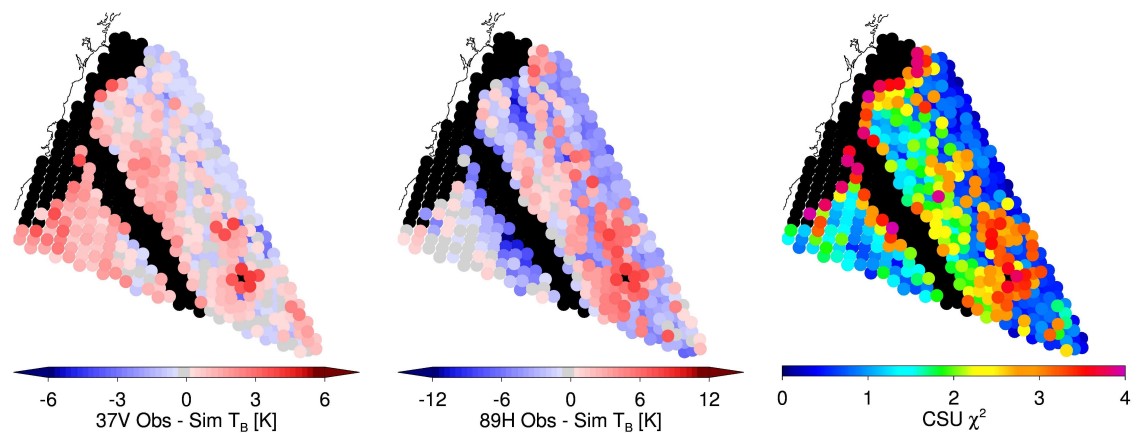

**Figure 11.** The same scene as Fig. 8, showing the difference of observed and simulated radiances, specifically at 37V (left) and 89H (middle) channels, and quality of fit (right). Black signifies pixels that did not converge.

## 6.3 Implications

This study demonstrates that explicit forward modelling of warm rain in a passive-only variational algorithm can indeed work if constructed and constrained properly. Observed radiances can be matched to modelled radiances successfully in a selection of raining scenes if DSD variability is taken into account. Figure 11 shows observed minus simulated radiances for two GMI channels, with little difference exhibited between raining and non-raining cloudy pixels. Similarly, the bottom panel of Fig. 12 demonstrates that the 1DVAR realistically simulates all 13 GMI channels in raining conditions globally, typically within 2

to $3\,\mathrm{K}$ for the average channel without strong regional dependence. Though all-sky radiance assimilation is not a directly


comparable problem, this level of agreement with observed radiances has implications for how all-sky DA schemes could better match radiances in raining conditions.

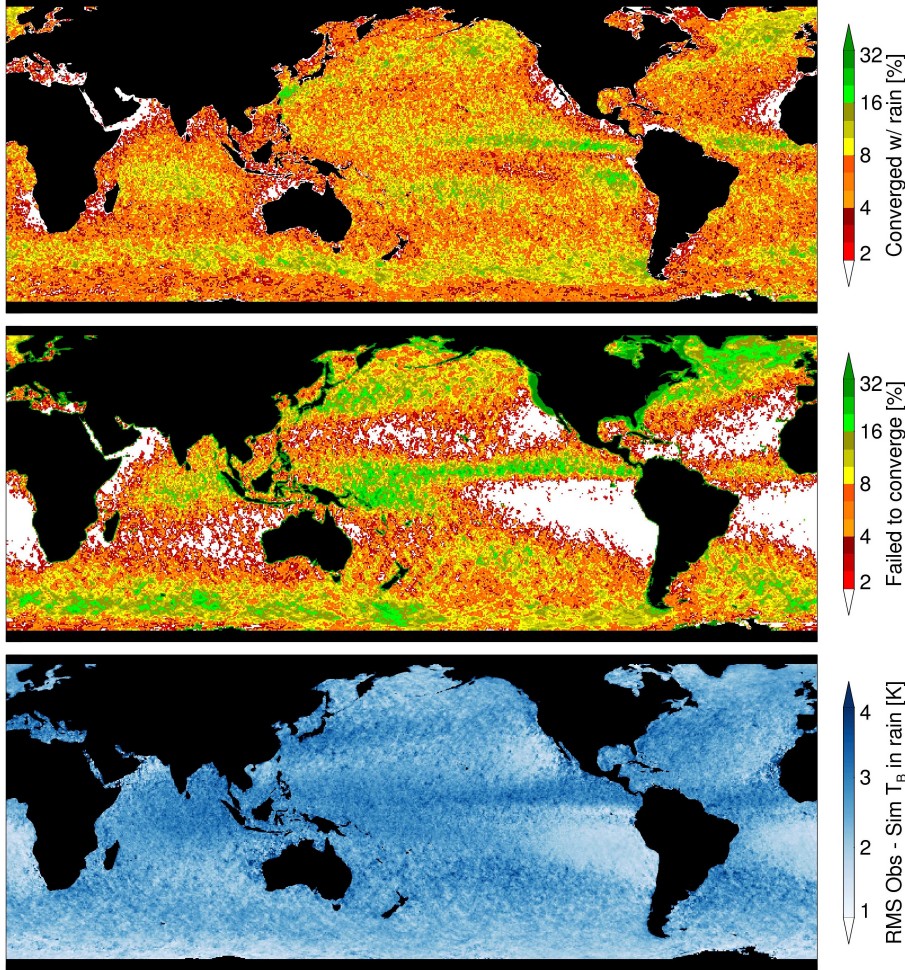

**Figure 12.** One year of GMI 1DVAR retrievals gridded at $0.5°$ resolution. The panels show: frequency of pixels with a non-zero rain rate (top), frequency of non-convergent pixels (middle), and the average root mean squared (RMS) errors between observed and simulated $T_B$ over all 13 channels for raining retrievals only (bottom).

Figures 12 and 13 offer a global, more climatological view of the warm rain retrieval, using the same 12 months of retrievals located within the DPR MS swath from the analysis in Section 5.3. The frequency of converged 1DVAR raining retrievals

5   lies between 2-10% for much of the global ocean. This can be contrasted with the frequency of non-convergent retrievals to approximate the relative frequency of warm rain versus all precipitation. However, while it bears similarity to the map of GPROF rain rates in Fig. 13, the retrieval can fail for reasons other than precipitation not represented by the forward model. For instance, much of the United States' coast exhibits a high frequency of non-convergent retrievals. This is a function of radio



frequency interference at 19 GHz, a documented issue for GMI radiances in that region (Draper and Newell, 2015). Similarly, the algorithm relies on a $\chi^2$ threshold for output, and thus the relative frequency of retrieved warm rain will vary if using different $\chi^2$ thresholds.

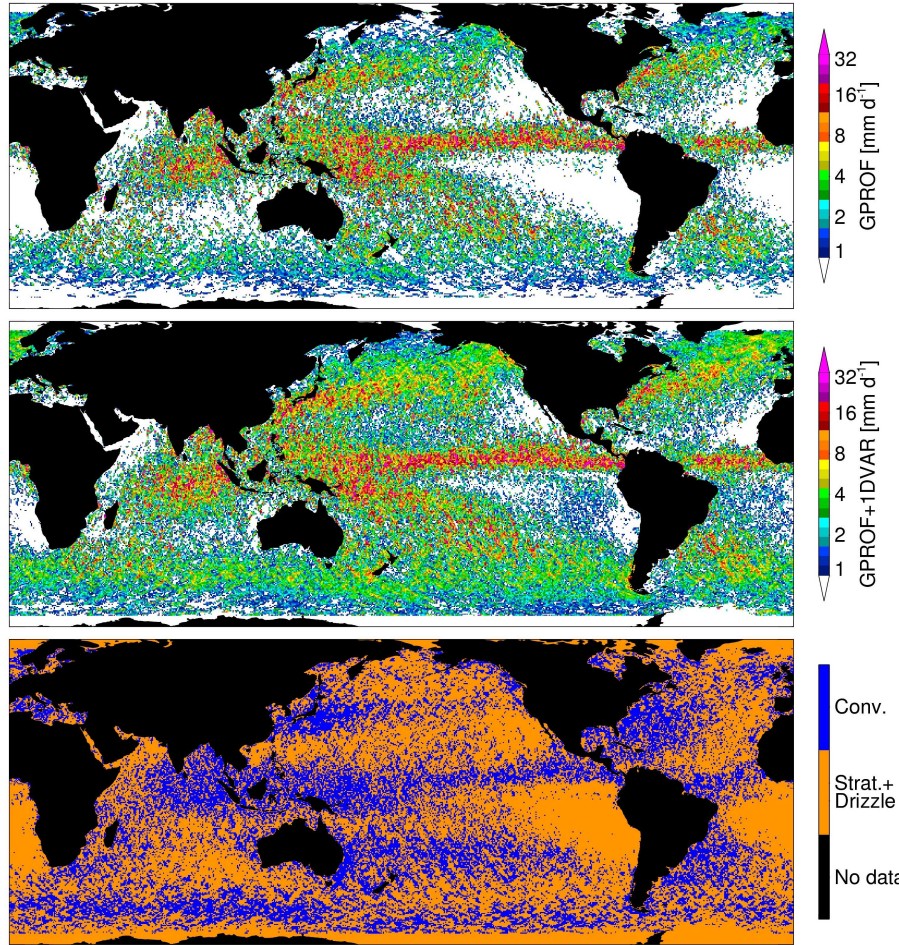

**Figure 13.** One year of GMI 1DVAR retrievals gridded at 0.5° resolution. The top panel shows GPROF mean rain rates from GMI. The middle panel shows a combination of GPROF plus 1DVAR-derived rain rates—the 1DVAR solution supersedes GPROF wherever it converged in rain and non-raining conditions. The bottom panel shows the dominant modes of 1DVAR precipitation retrieved.

An unresolved question is how to reconcile the differences between CloudSat-derived and GPM-derived precipitation distributions over the global oceans (Skofronick-Jackson et al., 2017, Fig. 5). GPROF and DPR observe less precipitation accumulation over the high latitude oceans and stratocumulus regions, for instance, a function of their limited sensitivities and sensor resolution (Behrangi et al., 2012). To probe this question, the 1DVAR rain rates were added to GPROF to ascertain the impact on the global rain distribution, seen in Fig. 13. For all raining and non-raining pixels where the 1DVAR converged, GPROF values were supplanted by the 1DVAR rain rate and the averages recomputed. To be conservative, only 1DVAR retrievals with



a fit to observations within prescribed errors ($\chi^2 < 1$) were included. This results in more rain just about everywhere over the global oceans, but it especially enhances accumulated rain in many regions where disagreements between CloudSat and GPM are strongest.

It is not surprising that 1DVAR-derived rainfall brings GMI retrieval totals more in line with those of CloudSat due to the algorithm's reliance on CloudSat data for drizzle onset thresholds. However, the relative simplicity of the 1DVAR's forward model and rainfall rate calculation—especially for drizzle—means that these results should be treated with caution. This should be considered as a naive estimate with potentially strong regional biases. Greater physical understanding and dedicated work into rain rates from drizzle in different regions would be needed to provide such an estimate with confidence.

For the stated reasons, a Bayesian retrieval such as GPROF still has advantages over a variational scheme for operational global products of precipitation. But it is conjectured that a blended Bayesian/variational approach may be preferable for current generation radiometers, as warm rain's relatively small signal to noise can be ascertained better by a variational algorithm while anything beyond warm rain is currently better handled via Bayesian integration. Hyperspectral passive microwave sensors could provide better observational constraints for a variational algorithm in the future (Birman et al., 2017), but current sensors' information content limitations dictate that sensing precipitation from a passive satellite platform requires many compromises yet.

## 7   Summary and conclusions

This study has explored the feasibility of extending variational passive microwave retrievals from non-raining/non-scattering regimes into the simplest precipitation regimes to forward model, namely oceanic warm rain and drizzle. This extension of a 1DVAR retrieval was accomplished via use of CloudSat-derived a priori information for hydrometeor profiles and drizzle onset, combined with a novel treatment of forward model errors caused by DSD assumptions. This augmentation of the retrieval described by Duncan and Kummerow (2016) was applied to a year of GMI data to assess its performance. Proofs of concept in Sect. 5 demonstrated that the variational retrieval can add information on precipitation in selected scenes. This was judged relative to an operational algorithm using Bayesian integration and a case in which drops exist between the sensitivity limits of the CloudSat and GPM radars (Fig. 7), results that are in line with theory. Limitations and sensitivities of the experimental retrieval were discussed in Sect. 6, with the drizzle onset threshold the key sensitivity. Limitations of the approach include the crude forward model and the ambiguity of assigning drizzle or warm rain. The transition from cloud to drizzle and warm rain is continuous, reflected in a continuum of $T_B$ response, and delineation between raining or non-raining states has to rely on quality of fit metrics to collapse this into algorithmic rules.

It is concluded that a variational retrieval can add information relative to operational precipitation products, albeit in limited circumstances. Treatment of correlated forward model errors, especially those caused by DSD assumptions, is important— analysis herein shows that errors vary strongly, depending on frequency, columnar rainwater, and meteorological regime (Fig. 5, Table 2). Collapsing the DSD variability to a binary classification was effective enough to permit convergence in a variety of regimes and simulate radiances with fidelity (Figs. 11, 12), an approach that can be adapted as data on global DSD variability





improves. The rain rate estimates proffered by this experimental retrieval are admittedly simplistic due to beam-filling and evaporation not being considered, and it remains to be seen whether such an approach can be extended to other types of precipitation. However, it is conjectured that the variational approach described here could be useful for future operational precipitation retrievals and radiance assimilation schemes, a way to maximise the information currently available from passive

5  microwave sensors.

*Code and data availability.*  The retrieval code referenced in this study is available (doi:10.5281/zenodo.1098212) along with sample output files.

*Competing interests.*  The authors declare no competing interests or conflicts.

*Acknowledgements.*  This study was primarily supported under NASA AMSR2 grant NNX13AN44G at Colorado State University, with

10  some additional support from the Swedish National Space Board at Chalmers University of Technology. The GEOS5 data used in this study have been provided by the Global Modeling and Assimilation Office (GMAO) at NASA Goddard Space Flight Center. Thanks as well to the NASA Precipitation Processing System, GPM Ground Validation, and CloudSat Data Processing Center teams for data access and support. Thanks as well to Patrick Eriksson at Chalmers for feedback and the anonymous reviewers for constructive criticisms.



## Appendix A: Appendix figures

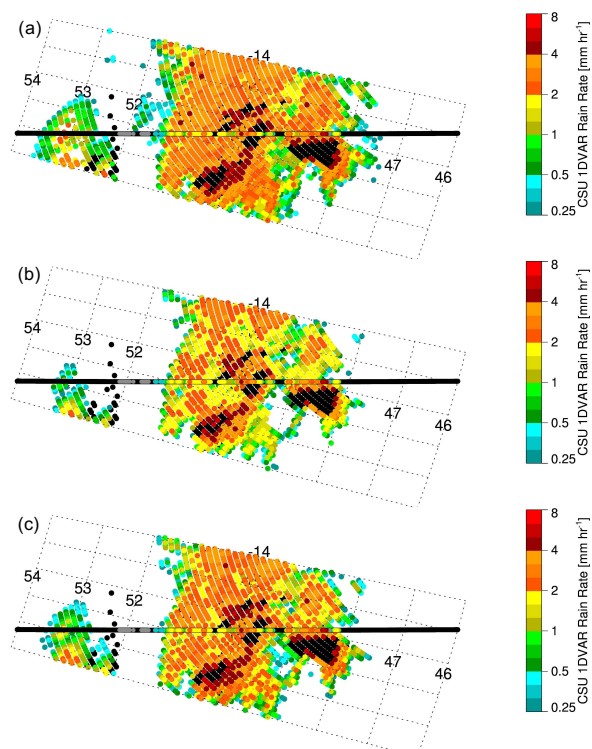

**Figure A1.** Sensitivity tests following Fig. 7. The panels show retrieved rain rates from the 1DVAR in cases with (a) LWP drizzle onset threshold decreased $50\,\mathrm{g\,m^{-2}}$, (b) increased $50\,\mathrm{g\,m^{-2}}$, and (c) using GPROF columnar rain water for the a priori state.





(a)       (b)       (c)

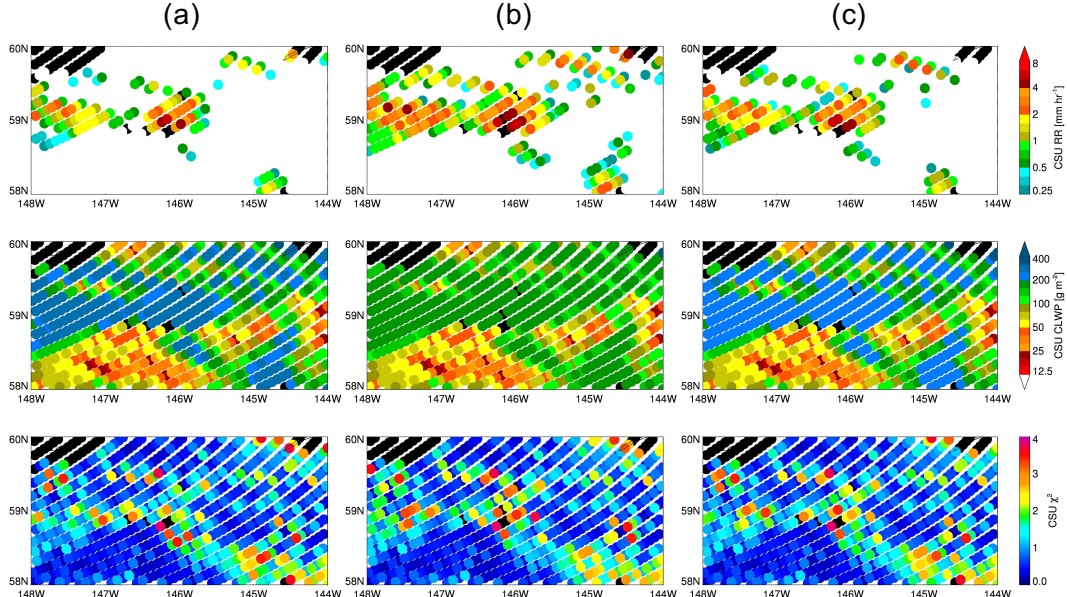

**Figure A2.** Sensitivity tests following Fig. 10. The panels show retrieved rain rates from the 1DVAR in cases with (a) LWP drizzle onset threshold decreased $50\,\mathrm{g\,m^{-2}}$, (b) increased $50\,\mathrm{g\,m^{-2}}$, and (c) using GPROF columnar rain water for the a priori state.

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
