# Peer review of "Towards variational retrieval of warm rain from passive microwave observations"

_Atmospheric Measurement Techniques, 2018_

## Referee Comment (RC1) · Anonymous Referee #1 · 30 May 2018

**General comments** This manuscript details a variational methodology to retrieve warm rain using passive sensors. The algorithm extracts hydrometeor and precipitation onset data from active sensors based on regime, coupled with radiative transfer models and disdrometer observations, to improve detection of raining warm clouds. This proof-of-concept provides promising results towards expanding the precipitation regimes that passive sensors can retrieve. Research presented in this manuscript is novel and reaches substantial conclusions. In particular, the care in the treatment of error provides readers with a well-informed and logical presentation of the algorithm. A few specific comments are offered as a guide to improve the manuscript.

**Specific comments**

[Figure]

1. You describe a lack of evaporation model, but included in the 2C-Rain-Profile algorithm is an evaporation model (Lebsock and L'Ecuyer, 2011, Sec. 3.4.2). Therefore, it may be difficult for a reader to discern from the text where exactly your algorithm deviates from the evaporation included in the CloudSat algorithm that information is derived from. Could you please clarify this a bit when making caveats that no evaporation model is included in the algorithm (e.g. P14L19-20, P26L2)?

2. Based on discussion and results from Figures 9 and 10, the goodness of fit provides a cursory estimate for trustworthiness of the retrieval. Figures 7 and 8 could be better served by including the goodness of fit estimates as there are locations north of 53° with 1DVAR rain retrievals but no discernable reflectivities from CloudSat in the lowest few kilometers of the troposphere. Those estimates coupled with the visual information of the reflectivity vertical profiles would provide further backing to that claim.

3. Along the lines of Point 2, was there any sensitivity analysis performed on the -8 K threshold used for higher frequency scattering by ice detection (P13L17) in the retrieval?

4. In Section 5.3, rain rates are excluded below 0.2 mm/hr for statistical calculations. It would be good to at least get an idea of the "distribution's tail" that exists from 0.2 mm/hr to e.g. 0.01 mm/hr. Though this manuscript is not intended to be an operational algorithm description and validation, it would be useful for the reader to understand the relative performance of 1DVAR compared to DPR or PR estimates in this regime specifically (which, for the latter at least, is essentially null in Berg, L'Ecuyer, and Haynes (2010), Figure 3).

**Technical corrections**

[Figure]

1. P3L25-26: "providing the best information content for sensing precipitation of any extant passive sensor": this statement should probably have a citation or be restated as it currently conveys a quantitative information content estimate.

2. P4L24: This should perhaps be rephrased to the National Weather Service operates.

3. P7L7: Please specify in the text the two years of CloudSat data.

4. Figure 3: Please include units on the figure.

5. P28-31: Some references are missing DOIs. P30L21 has a malformed DOI.

**References**

- Berg, W. S., L'Ecuyer, T. S., and Haynes, J. M.: The distribution of rainfall over oceans from spaceborne radars, J. Appl. Met. Clim., 49, 535–543, https://doi.org/10.1175/2009JAMC2330.1, 2010.

- Lebsock, M. and L'Ecuyer, T. S.: The retrieval of warm rain from CloudSat, J. Geophys. Res., 116, https://doi.org/10.1029/2011JD016076, 2011.

---

## Referee Comment (RC2) · Anonymous Referee #2 · 1 Jun 2018

This is an excellent study which helps understand the discrepancy between GPM/GPROF and Cloudsat precipitation retrievals, points a way forward for better retrievals of warm rain using passive microwave observations, and finally introduces some innovative and useful techniques, such as correlated observation errors that account for the unknown dropsize distribution. In places the methods need to be explained more precisely and comprehensively, and there are a few other minor comments, but otherwise this manuscript is close to being ready for publication. Many of the comments focus on the a-priori and its error covariance matrix, details of which are essential for understanding the basic characer of the retrieval algorithm, i.e. how closely it attempts to fit observations, how much of the retrieval is determined by the a-priori, and whether or not the retrieval is truly optimal.

[Figure]

Minor comments

1) CloudSat has trouble seeing the lowest km, as acknowledged on page 16, lines 1-2. But the proposed retrieval makes use of CloudSat rain profiles to generate a relationship between RWP and surface rain rate (page 13, lines 6-10). Surely the proposed retrieval is thus also affected by these errors? In any case, it would be good to explain briefly how the CloudSat rain rates are generated and hence what assumptions are being made.

2) On page 11: "Figure 5 translates the simple model containing in situ DSD data into error covariances matrices usable by the retrieval, via Eq. 2". This is not stricly true. Equation 2 gives only the derivation of the diagonal elements of the covariance matrix. Please also define how the off-diagonal elements are computed.

3) On page 11, this sentence is not clear: "To apply these analyses of in situ data as realistically as possible, the errors and DSD assumptions derived from extratropical and tropical sites are dependent on the observed latitude." How is this done? How does this relate to what is shown in figure 5? What is the relationship between "extratropical and tropical sites" and "dependent on observed latitude"? All needs to be explained in a little more detail, or perhaps rephrased to be clearer.

4) On page 11, line 14 (and corrspondingly, in Figure 6) please define the criteria for non-convergence used in the non-raining algorithm. More generally the phrases "not converged" and the notation "$\chi^2 < 1$" crop up throughout the paper and it needs to be clear whether these are one and the same, or whether "not converged" has any further meaning. For example, depending on the solver being used, there can be many other failure modes for a variational retrieval.

5) Page 11, line 27-28 the observation error covariance matrix "$S_y$ for the non-raining retrieval is the same as that given by Duncan and Kummerow (2016)". The reader needs a few sentences to describe how this is constructed. It should not be necessary to have to go to the cited paper.

[Figure]

6) Further, the a-priori error covariance matrix $S_a$ for the non-raining retrieval needs a brief description and quantification (see point 9).

7) The $\chi^2$ statistic of equation 3 is inexact, with a number of consequences. The true Chi squared test for the consistency of an optimal variational retrieval given the specified error covariance matrices (see Rodgers, 2000, equation 12.9, as cited in the manuscript) would replace the observation error covariance matrix $S_y$ in this manuscript's approximate Chi squared test with $S_y(KS_aK^t + S_y)^{-1}S_y$. Here $K$ (Rodgers' notation) would correspond to the Jacobian of the observation operator (the derivatives of $f(x, b)$ in this manuscript) and $S_a$ the error covariance of the a-priori. The issues are, first, that it is confusing to label a test $\chi^2$ when it is not the true test - and a word of explanation could be given - why did the authors choose this less exact formulation? Second, when the a-priori error covariance in observation space is small relative to the observation errors (i.e. $KS_aK^t > S_y$) the manuscript's inexact Chi squared becomes a poor measure of convergence - broadly, the retrieval will not get much closer to the observations, but this is OK because it is consistent with the specified errors. Third, the true Chi squared test could have given a powerful statistical tool for assessing the validity of the assumed error covariance matrices, which would have been very useful for understanding the retrieval methodology.

8) In section 4.3, the source of the a-priori for the warm rain retrieval does not seem to be described. This would be helpful for its own sake, but also to support section 6.2 which describes what happens when the a-priori is changed - but with respect to what?

9) In section 4.3, the a-priori error covariance matrix is not introduced, but it is essential for understanding the character of the retrieval algorithm (essentially what are the relative sizes of $KS_aK^t$ and $S_y$, i.e. how closely does it attempt to fit the observations). For example, the description, "Even with four variables, the a priori errors on LWP and PC1 of water vapour are decreased" is not precise enough, especially since there is currently no quantification of the non-raining error covariance matrix (see point 6)

10) In table 2, does the first row of data correspond to $S_y$ for the non-raining retrieval? If so it would be helpful to state this explicitly. If not, this would be a good place to illustrate it.

11) Page 14, line 16: "if neither converges" - please be more precise on how this is defined.

Technical comments

1) Figure 5 caption makes use of the notation $S_y$ before it has been introduced.
* * *

---

## Referee Comment (RC3) · Anonymous Referee #3 · 12 Jun 2018

This study examines the feasibility of using passive-only microwave satellite measurements to retrieve drizzle and rain in warm oceanic clouds. It's a topic that has been largely unexplored and the authors have made, in my opinion, a convincing case that it can be done but with caveats. Their results have confirmed my suspicions about the DPR missing a good portion of drizzle and light rain. Some argue that it doesn't matter since the amount of liquid mass associated with warm rain and drizzle is insignificant from a hydrological point of view. However, studies have shown that drizzle production in stratocumulus clouds has a significant impact on cloud albedo, which has implications for global climate.

The authors also have a firm understanding of the challenges involved and have thoroughly considered all sources of error and the method's limitations. Dynamical adjustment of the observation error covariance matrix in the context of rain drop size distributions is perhaps the most novel part of the study.

Specific comments:

Page 3, lines 1-9: As pointed the authors point out, there are important implications for this work in all-sky radiance data assimilation (DA). It is also important to note that aside from characterization of the observation and forward model errors, there are also limitations in the simulation of cloud fields in NWP models. Nearly all NWP models contain crude microphysics, not allowing the DA system to take full advantage of the information content provided by passive microwave measurements. Section 3.1: Partitioning the drizzle onset by other environmental parameters (TPW/SST) is a good way to deal with the fact that methods which use a single LWP threshold in all conditions (e.g., Wentz and Spencer 1998) do not work very well for warm clouds (Greenwald et al. 2018). Section 6.2: For me, the bottom line in the usefulness of a 1DVar retrieval is knowing whether it relies too much on the a priori information. One way to quantify how much the 1Dvar retrievals depend on this information is to compute the matrix Da = 1-A, where A = Dy K (K is the kernel matrix), and where Dy = Sa KˆT(K Sa KˆT + Sy ) ˆ(-1), which uses the same notation as in eq (3). Da provides a measure of the contributions of the combined measurements relative to the a prior information. It varies between 0 and 1, where 0 means the retrieval is relying completely on the combined measurements and 1 means the retrieval relies completely on the a priori state.

A 1DVar retrieval also provides a means of computing the retrieval error, which may also be useful. Have the authors considered computing this error? Other comments and corrections:

Page 2, lines 16-17: It's not clear to me why only a 1DVar retrieval and not other retrieval techniques should be "sensitive to rainfall below the detectability of the DPR." Is there a past study that demonstrates this comparative sensitivity?

Page 2, lines 20-27: I feel that the description of the Bayesian method is a bit over-simplified. A very good overview of the method is described in Evans et al. (1995; JAM).

Page 3, line 13: " For a rain retrieval . . ."

Page 4, lines 6-7: What spatial resolution are the brightness temperatures convolved to?

Page 5, line 1: I don't necessarily agree with the general statement that "passive radiances at typical frequencies contain almost no information on the vertical structure of hydrometeors." This is probably true for shallow warm clouds, but not for deep convective clouds. Studies have shown there is in fact some vertical information available (see, e.g., Smith et al. 1992; Evans et al. 1995).

Page 5, first paragraph of section 3.1: This paragraph contains well known information that could be removed.

Page 5, Figure 1: You might consider using color instead of grayscale.

Page 6, lines 1-2: I would mention somewhere that the retrieved LWP is actually the total LWP, that is, rain water path plus cloud liquid water path.

Page 6, line 4: I'm assuming the "observations" are the GMI observations?

Page 6, line 5: This part of the sentence is a bit confusing. What do you mean by "precipitation frequency?"

Page 15, line 19: "excepting" should be "except"

Page 16, lines 19-20: The sentence beginning with "The few pixels raining hardest . . ." sounds awkward. Consider rewriting as "The few pixels containing the largest rain rates . . ."

Page 18, last sentence: The meaning of the sentence beginning with "Here a threshold

of . . .” is unclear. Would it be possible to rewrite this sentence?

Page 24, line 1: What about radio interference at 10 GHz?

Page 25, line 15: Should "yet" be at the end of the sentence?

——————————————————

---

## Author Comment (AC1) · 26 Jun 2018

**Response to Reviewers, AMT-2018-96, "Towards variational retrieval of warm rain from passive microwave observations"**

David Ian Duncan et al.

**General response**

Thanks to all three reviewers for the helpful comments and suggestions offered to improve the manuscript. The reviewer comments follow, in italics, with the authors' responses following each along with quoted sections of text whenever the manuscript has been changed.

**1 Reviewer 1**

**1.1 Specific comments**

*1. You describe a lack of evaporation model, but included in the 2C-Rain-Profile algorithm is an evaporation model (Lebsock and L'Ecuyer, 2011, Sec. 3.4.2). Therefore, it may be difficult for a reader to discern from the text where exactly your algorithm deviates from the evaporation included in the CloudSat algorithm that information is derived from. Could you please clarify this a bit when making caveats that no evaporation model is included in the algorithm (e.g. P14L19-20, P26L2)?*

1. The text has been amended to make clear that some evaporation may be implicit due to the shape of the CloudSat-derived profile shapes: "...other than that implicit in the shape of the RWC profile (Fig. 2)."

*2. Based on discussion and results from Figures 9 and 10, the goodness of fit provides a cursory estimate for trustworthiness of the retrieval. Figures 7 and 8 could be better served by including the goodness of fit estimates as there are locations north of 53 with 1DVAR rain retrievals but no discernable reflectivities from CloudSat in the lowest few kilometers of the troposphere. Those estimates coupled with the visual information of the reflectivity vertical profiles would provide further backing to that claim.*

2. This is a good suggestion, and one that we entertained, but for the figures being quite busy already. Below is a sample of adding the quality of fit into the line plot of Fig. 7, which gives an idea of how well the retrieval converged for the pixels in question. If this is felt to be a significant aid to interpretation of the figure then it can be added to one or both of the plots, but in the authors' estimation those particular figures are already quite densely packed with information. The light rain in question near 53N appears to be supercooled water primarily, which is perhaps why the forward model does relatively poorly fitting the Tbs in that area. To emphasize, this is why a strong ($\chi^2 < 1$) threshold is used in the final analysis with GPROF, and thus many of those pixels would not supplant the GPROF result.

[Figure]

**Figure 1.** The second panel from Fig. 7, with quality of fit included as a dotted line (y-axis is shared, with $\chi^2$ unit-less).

*3. Along the lines of Point 2, was there any sensitivity analysis performed on the -8 K threshold used for higher frequency scattering by ice detection (P13L17) in the retrieval?*

Some ad hoc sensitivity testing was done with this threshold when initially testing the algorithm, but nothing formal. It was found that a combination of the 166GHz and 183±7 was best for looking for an ice scattering signature since the 183±3 channel usually had a lesser response due to water vapor absorption. Values like -10 and -6K were used initially, but one tended to toss aside pixels in which convergence could be reached while the other caused too many non-warm rain events to have retrievals attempted. The use of a threshold is mainly to help save some processing time by not attempting retrievals which are almost assuredly outside the scope of the forward model.

*4. In Section 5.3, rain rates are excluded below 0.2 mm/hr for statistical calculations. It would be good to at least get an idea of the "distribution's tail" that exists from 0.2 mm/hr to e.g. 0.01 mm/hr. Though this manuscript is not intended to be an operational algorithm description and validation, it would be useful for the reader to understand the relative performance of 1DVAR compared to DPR or PR estimates in this regime specifically (which, for the latter at least, is essentially null in Berg, L'Ecuyer, and Haynes (2010), Figure 3).*

4. The main reason this was avoided was not because the 1DVAR rain rate distribution has a long tail, but rather the distribution of DPR averaged into the GMI footprint, as 81 DPR pixels were weighted to determine the rain rate for the GMI FOV. This makes comparison with the 1DVAR rain rates, and determining whether they agree that it is raining or not, difficult if a threshold is not used. For example, the table below gives the percentage of GMI pixels from a given month for which the 1DVAR and DPR rain rates are above certain cutoff values. The number of 1DVAR raining pixels does not change markedly depending on cutoff value, but the DPR-based result does change quite a lot.

**Table 1.** Percentage of GMI pixels in Sept 2014 with rain rates above some given cutoffs.

| Cutoff [$mmhr^{-1}$] | 1DVAR-only | DPR-only | Either |
|:---:|:---:|:---:|:---:|
| 0.005 | 7.30 | 15.4 | 27.3 |
| 0.05 | 7.02 | 9.32 | 20.9 |
| 0.20 | 6.19 | 5.34 | 15.9 |

**1.2 Technical comments**

*1. P3L25-26: "providing the best information content for sensing precipitation of any extant passive sensor": this statement should probably have a citation or be restated as it currently conveys a quantitative information content estimate.*

This has been modified to use weaker language, as the initial statement was indeed too strong without an appropriate citation: "...providing information content for sensing liquid hydrometeors and some frozen hydrometeors (Birman et al., 2017)"

*2. P4L24: This should perhaps be rephrased to the National Weather Service operates.*

Done: "The National Weather Service operates a dual-pol radar..."

*3. P7L7: Please specify in the text the two years of CloudSat data.*

This has been added: "...warm rain, 2014 and 2015."

*4. Figure 3: Please include units on the figure.*

Done

*5. P28-31: Some references are missing DOIs. P30L21 has a malformed DOI.*

References have been updated to all contain DOIs.

**2 Review 2**

**2.1 Minor comments**

*1) CloudSat has trouble seeing the lowest km, as acknowledged on page 16, lines 1-2. But the proposed retrieval makes use of CloudSat rain profiles to generate a relationship between RWP and surface rain rate (page 13, lines 6-10). Surely the proposed retrieval is thus also affected by these errors? In any case, it would be good to explain briefly how the CloudSat rain rates are generated and hence what assumptions are being made.*

1) Yes, the drizzle rate calculation is indeed affected by these errors. A quick explanation of how the Lebsock algorithm calculates rain rates has been added to description of 2C-RAIN-PROFILE: "The rain rate is calculated via a Z-R relationship that is dependent on cloud type, with lower rain rates primarily a function of near-surface reflectivity while higher rain rates are more a function of path integrated attenuation (Lebsock and L'Ecuyer, 2011, Fig. 6)."

*2) On page 11: "Figure 5 translates the simple model containing in situ DSD data into error covariances matrices usable by the retrieval, via Eq. 2". This is not stricly true. Equation 2 gives only the derivation of the diagonal elements of the covariance matrix. Please also define how the off-diagonal elements are computed.*

2) The analysis method used to extract the covariances from the DSD forward model simulations is a bit convoluted so as to permit the formulation of dynamically scaling covariances. In practice, this meant calculating error variances and their respective correlations for different quantiles of the distribution. To formulate the covariance matrix each time, the algorithm interpolates the sigmas and correlation coefficients, then computes the full covariance matrix from these constituents. To reflect this without getting too bogged down in the details, the text has been modified thus: "...via Eq. 4 and the attendant correlation coefficients between channels' errors."

*3) On page 11, this sentence is not clear: "To apply these analyses of in situ data as realistically as possible, the errors and DSD assumptions derived from extratropical and tropical sites are dependent on the observed latitude." How is this done? How does this relate to what is shown in figure 5? What is the relationship between "extratropical and tropical sites" and "dependent on observed latitude"? All needs to be explained in a little more detail, or perhaps rephrased to be clearer.*

3) This has been rephrased and expanded to clarify: "To apply these analyses of in situ data as realistically as possible, the errors and DSD assumptions derived from extratropical and tropical sites are treated separately. The errors and assumptions applied depend on the observed latitude, with 30° latitude acting as the separator. Fig. 5 displays errors using the extratropical sites' data."

*4) On page 11, line 14 (and corrspondingly, in Figure 6) please define the criteria for non-convergence used in the non-raining algorithm. More generally the phrases "not converged" and the notation "X2 < 1" crop up throughout the paper and it needs to be clear whether these are one and the same, or whether "not converged" has any further meaning. For example, depending on the solver being used, there can be many other failure modes for a variational retrieval.*

4) A sentence has been added to the opening paragraph of Section 4: "Non-convergence for each stage is defined by either failure to converge within 10 iterations or very poor fit ($\chi^2 > 4.0$)."

*5) Page 11, line 27-28 the observation error covariance matrix "Sy for the non-raining retrieval is the same as that given by Duncan and Kummerow (2016)". The reader needs a few sentences to describe how this is constructed. It should not be necessary to have to go to the cited paper.*

5) The following text has been added: "The non-raining observation error covariances account for misplacement of cloud and water vapor in the atmospheric column, as well as surface pressure, wind direction, salinity, and emissivity model errors; the channel variances for non-raining cases are given in Table 2."

*6) Further, the a-priori error covariance matrix Sa for the non-raining retrieval needs a brief description and quantification (see point 9).*

6) In the next paragraph we have added the following. If readers are interested in the specific values used for each variable, these are visible in the code which is available through the archived copy on Zenodo. "A priori covariances for wind speed and water vapour were derived from reanalysis data; as reanalysis cloud water is not representative, only the covariance between LWP and the first PC of water vapour is included."

*7) The X2 statistic of equation 3 is inexact, with a number of consequences. The true Chi squared test for the consistency of an optimal variational retrieval given the specified error covariance matrices (see Rodgers, 2000, equation 12.9, as cited in the manuscript) would replace the observation error covariance matrix Sy in this manuscript's approximate Chi squared test with Sy(KSaKt + Sy)-1Sy. Here K (Rodgers' notation) would correspond to the Jacobian of the observation operator (the derivatives of f(x,b) in this manuscript) and Sa the error covariance of the a-priori. The issues are, first, that it is confusing to label a test X2 when it is not the true test - and a word of explanation could be given - why did the authors choose this less exact formulation? Second, when the a-priori error covariance in observation space is small relative to the observation errors (i.e. KSaKt > Sy) the manuscript's inexact Chi squared becomes a poor measure of convergence - broadly, the retrieval will not get much closer to the observations, but this is OK because it is consistent with the specified errors. Third, the true Chi*

*squared test could have given a powerful statistical tool for assessing the validity of the assumed error covariance matrices, which would have been very useful for understanding the retrieval methodology.*

7) This comment prompted a great deal of discussion. Taking the points in order:

1. We were more interested in the fit to $T_B$s alone, hence why it is referred to as the fit metric. But you are right that this is confusing, and the text now tries to clarify this: "The fit metric ($\chi^2$) is normalised by the number of channels used, and indicates the quality of fit between the retrieved state's simulated $T_B$s and those observed. Note that this is not a true $\chi^2$ test (Rodgers, 2000, Eq. 12.9), but instead used to gauge fit to the observations alone."

2. This point should not be a concern for the retrieval because $\chi^2$ as defined is not used to determine convergence. In fact, Eq. 5.33 from Rodgers is used ($D_i^2 << m$), a formulation that requires $S_{\delta y}$ in its calculation. Thus computation of the 'true' $\chi^2$ would be trivial for us to perform for future development of the 1DVAR algorithm.

3. We agree that use of the 'true' $\chi^2$ test could be interesting to compare against the fit to $T_B$s $[(y - f(x))^T S_y^{-1}(y - f(x))]$ to examine the validity of our error assumptions, though this is currently outside the scope of this study on an experimental retrieval.

*8) In section 4.3, the source of the a-priori for the warm rain retrieval does not seem to be described. This would be helpful for its own sake, but also to support section 6.2 which describes what happens when the a-priori is changed - but with respect to what?*

8) The following has been amended/added after the first paragraph of the section, and made a new paragraph to aid readability. Specifics about the PCs of water vapour, RWC, and PIWC are deemed not useful, as there is no easy physical interpretation for the reduced-space and unit-less variables. Qualitatively, the fact that water vapour and cloud water variables have to be more tightly constrained (i.e. 40% of the variance used in the non-raining retrieval, which comes from ERA-Interim statistics) is perhaps the most salient for readers.

"Even with four variables, the a priori errors on LWP and PC1 of water vapour are decreased, to $10\,\mathrm{g\,m^{-2}}$ and 60% smaller, respectively, so as to discourage unphysical behaviour in the retrieval, with the prior for LWP coming from the non-raining retrieval. These tighter constraints help to avoid a tendency of the retrieval to push humidity and cloud water to very high levels in some cases. A priori errors on the profiles of RWC and PIWC come from global CloudSat statistics that produced Fig. 2. Raining scenes can exhibit 1.5-3.5 degrees of freedom for signal (DFS) given these four retrieved parameters, indicating that even with four variables the problem is information-limited."

*9) In section 4.3, the a-priori error covariance matrix is not introduced, but it is essential for understanding the character of the retrieval algorithm (essentially what are the relative sizes of KSaKt and Sy, i.e. how closely does it attempt to fit the observations). For example, the description, "Even with four variables, the a priori errors on LWP and PC1 of water vapour are decreased" is not precise enough, especially since there is currently no quantification of the non-raining error covariance matrix (see point 6)*

9) This is addressed in response to 8) above. But in addition, the suggestion that comparison of $K S_a K^T$ and $S_y$ would be worthwhile is well taken, and will inform future development. However, since the Jacobian ($K$) and $S_y$ change scene by scene

as well as iteration to iteration, it is hard to think of a good way to compare these matrices in a systematic way that would shed light on the retrieval behaviour rather than muddying the waters.

*10) In table 2, does the first row of data correspond to Sy for the non-raining retrieval? If so it would be helpful to state this explicitly. If not, this would be a good place to illustrate it.*

10) Yes, this is correct. The figure caption has been amended to state this explicitly: "The first line (RWP=0) shows the non-raining algorithm's error variances."

*11) Page 14, line 16: "if neither converges" - please be more precise on how this is defined.*

11) As with the clarification for the non-raining retrieval, comment 4) above, this is based on poor fit to $T_B$s or too many iterations. Non-convergence is thus treated the same for raining retrievals as for non-raining.

**2.2 Technical comments**

*1) Figure 5 caption makes use of the notation Sy before it has been introduced.*

To link this better with the rest of the manuscript, the following was added to the figure caption: "These error covariances make up $S_{y,rain}(RWP)$, a constituent of the total $S_y$ from Eq. 3, described later in Section 4.3."

**3 Review 3**

**3.1 Specific comments**

*Page 3, lines 1-9: As pointed the authors point out, there are important implications for this work in all-sky radiance data assimilation (DA). It is also important to note that aside from characterization of the observation and forward model errors, there are also limitations in the simulation of cloud fields in NWP models. Nearly all NWP models contain crude microphysics, not allowing the DA system to take full advantage of the information content provided by passive microwave measurements.*

This is a good point, and the following sentence has been added to this paragraph to highlight this: "NWP models often contain crude microphysics that limits their ability to accurately simulate clouds' radiative properties."

*Section 3.1: Partitioning the drizzle onset by other environmental parameters (TPW/SST) is a good way to deal with the fact that methods which use a single LWP threshold in all conditions (e.g., Wentz and Spencer 1998) do not work very well for warm clouds (Greenwald et al. 2018).*

Thanks for making us aware of that recent Greenwald et al. paper. To strengthen the motivation of regime-dependent drizzle onset the following sentence was added to this section: "A constant precipitation onset value can lead to pervasive systematic biases in cloud and rain retrievals (Greenwald et al., 2018)."

*Section 6.2: For me, the bottom line in the usefulness of a 1DVar retrieval is knowing whether it relies too much on the a priori information. One way to quantify how much the 1Dvar retrievals depend on this information is to compute the matrix Da = 1-A, where A = Dy K (K is the kernel matrix), and where Dy = Sa K^T(K Sa K^T + Sy ) ^(-1), which uses the same notation as in eq (3). Da provides a measure of the contributions of the combined measurements relative to the a prior information. It*

*varies between 0 and 1, where 0 means the retrieval is relying completely on the combined measurements and 1 means the retrieval relies completely on the a priori state.*

The trace of the $A$ matrix was used to examine the degrees of freedom for signal seen in the retrieval. This was not explored in much detail here due to space, but saw further discussion in Duncan (2017). As mentioned in Section 4.3, we saw anything from 1.5 to 3.5 DFS in raining pixels, signaling that the trace of $D_a$ can range from 0.5 to 2.5.

*A 1DVar retrieval also provides a means of computing the retrieval error, which may also be useful. Have the authors considered computing this error?*

Yes, and indeed the posterior error variances are output by the retrieval already, which can be seen in the sample output files available with the archived code on Zenodo. The tricky thing with analysing these posterior errors is that it requires conversion from the reduced space (PC1 of RWC, etc.) back into something meaningful (such as rain rate) to be of real use. This is far from trivial given the baked in assumptions about DSD and fall speed, for instance, and so thus far we have not attempted to do this for rainfall. These retrieval error statistics for non-raining retrievals were examined in depth in Duncan and Kummerow (2016).

*Page 2, lines 16-17: It's not clear to me why only a 1DVar retrieval and not other retrieval techniques should be "sensitive to rainfall below the detectability of the DPR." Is there a past study that demonstrates this comparative sensitivity?*

For clarification the word "variational" was removed from that sentence. It was not meant that only a VAR retrieval would have this sensitivity, rather that the operational Bayesian retrieval (GPROF) does not because it is limited to the sensitivity of the radar due to its reliance on DPR for its a priori. Other retrieval methods could potentially also demonstrate this sensitivity if they are treating the surface state more accurately than GPROF does.

*Page 2, lines 20-27: I feel that the description of the Bayesian method is a bit over- simplified. A very good overview of the method is described in Evans et al. (1995; JAM).*

This citation has been added to this section for an interested reader. The description of Bayesian retrieval is admittedly very brief, but it is not a focus of the paper while other papers contain fuller descriptions.

*Page 3, line 13: " For a rain retrieval . . ."*

Fixed

*Page 4, lines 6-7: What spatial resolution are the brightness temperatures convolved to?*

The L1CR product produces co-registered $T_B$s but these are not convolved, so the input radiances used are in fact all at the native resolution. This is certainly a limitation of the algorithm (touched upon tangentially as beam-filling in Section 6.2) but also keeps it consistent with the approach used by GPROF.

*Page 5, line 1: I don't necessarily agree with the general statement that "passive radiances at typical frequencies contain almost no information on the vertical structure of hydrometeors." This is probably true for shallow warm clouds, but not for deep convective clouds. Studies have shown there is in fact some vertical information available (see, e.g., Smith et al. 1992; Evans et al. 1995).*

This has been amended from "almost no information" to "little information"

*Page 5, first paragraph of section 3.1: This paragraph contains well known information that could be removed.*

The authors felt it worthwhile to keep this paragraph intact, as it motivates the method described and might not be common knowledge for all readers.

*Page 5, Figure 1: You might consider using color instead of grayscale.*

This figure has been modified to use the same colorscale as Fig. 2, which both keeps some consistency amongst the figures and allows differentiation of areas with no data, now shown in black. The updated figure and caption are given below.

[Figure]

**Figure 2.** Drizzle onset value of LWP, separated by SST and TPW. Regimes with little data are assigned the maximum value, $300\,\mathrm{g\,m^{-2}}$, in line with Wang et al. (2017), while regimes with no data are given in black.

*Page 6, lines 1-2: I would mention somewhere that the retrieved LWP is actually the total LWP, that is, rain water path plus cloud liquid water path.*

This particular instance of "LWP" has been amended to "total LWP"

*Page 6, line 4: I'm assuming the "observations" are the GMI observations?*

10    Yes, "observations" here has been modified to "GMI observations" to make this explicit.

*Page 6, line 5: This part of the sentence is a bit confusing. What do you mean by "precipitation frequency?"*

The precipitation frequency from CloudSat is the overall frequency of non-zero precipitation retrieved from the CloudSat PIA-based retrieval of Haynes et al., and the threshold was tuned until GMI exhibited the same overall precipitation frequency in each TPW/SST bin.

15    *Page 15, line 19: "excepting" should be "except"*

Fixed

*Page 16, lines 19-20: The sentence beginning with "The few pixels raining hardest..." sounds awkward. Consider rewriting as "The few pixels containing the largest rain rates..."*

Fixed

*Page 18, last sentence: The meaning of the sentence beginning with "Here a threshold of..." is unclear. Would it be possible to rewrite this sentence?*

In line with a comment from Reviewer 1, this has been modified to: "Here a threshold of $0.2\,\mathrm{mm\,h}^{-1}$ defines non-zero rain to avoid the distribution's tail that arises from averaging of DPR data into the GMI footprint." For further explanation see comment 4 from Reviewer 1 above.

*Page 24, line 1: What about radio interference at 10 GHz?*

It's true that RFI can exist at the GMI 10.65GHz band, but it is much more prevalent over land and thus not a big factor for our oceanic retrievals (Draper, 2018). This citation has also been updated to a newer Draper paper which is more comprehensive.

*Page 25, line 15: Should "yet" be at the end of the sentence?*

It's perhaps a bit on the poetic side, but it is grammatically correct.

**References**

Birman, C., Mahfouf, J. F., Milz, M., Mendrok, J., Buehler, S. A., and Brath, M.: Information content on hydrometeors from millimeter and sub-millimeter wavelengths, Tellus, Ser. A: Dyn. Meteorol. and Oceanogr., 69, 1271562, https://doi.org/10.1080/16000870.2016.1271562, 2017.

5  Draper, D. W.: Radio Frequency Environment for Earth-Observing Passive Microwave Imagers, IEEE J. Sel. Top. Appl. Rem. Sens., pp. 1–10, https://doi.org/10.1109/JSTARS.2018.2801019, 2018.

Duncan, D. I.: Exploring the Limits of Variational Passive Microwave Retrievals, Ph.D. thesis, Colorado State University, 2017.

Duncan, D. I. and Kummerow, C. D.: A 1DVAR retrieval applied to GMI: Algorithm description, validation, and sensitivities, J. Geophys. Res. Atmos., 121, 7415–7429, https://doi.org/10.1002/2016JD024808, 2016.

10  Greenwald, T. J., Bennartz, R., Lebsock, M., and Teixeira, J.: An Uncertainty Data Set for Passive Microwave Satellite Observations of Warm Cloud Liquid Water Path, J. Geophys. Res. Atmos., 123, 3668–3687, https://doi.org/10.1002/2017JD027638, 2018.

Lebsock, M. D. and L'Ecuyer, T. S.: The retrieval of warm rain from CloudSat, J. Geophys. Res. Atmos., 116, https://doi.org/10.1029/2011JD016076, 2011.

Rodgers, C. D.: Inverse methods for atmospheric sounding: Theory and practice, vol. 2, World Scientific, 2000.

15  Wang, Y., Chen, Y., Fu, Y., and Liu, G.: Identification of precipitation onset based on Cloudsat observations, J. Quant. Spectrosc. Radiat. Transfer, 188, 142 – 147, https://doi.org/10.1016/j.jqsrt.2016.06.028, 2017.